

# Distributed surface mass balance of an avalanche-fed glacier

Marin Kneib[1,2], Amaury Dehecq[1], Adrien Gilbert[1], Auguste Basset[1], Evan S. Miles[3,4], Guillaume Jouvet[5], Bruno Jourdain[1], Etienne Ducasse[1], Luc Beraud[1], Antoine Rabatel[1], Jérémie Mouginot[1,†], Guillem Carcanade[1], Olivier Laarman[1], Fanny Brun[1], Delphine Six[1]

[1]Institut des Géosciences de l'Environnement, Université Grenoble-Alpes, CNRS, IRD, Grenoble, 38400, France
[2]Department of Atmospheric and Cryospheric Sciences, University of Innsbruck, Innsbruck, 6020, Austria
[3]Glaciology and Geomorphodynamics Group, Department of Geography, University of Zurich, Zurich, 8057, Switzerland
[4]Mass Movements and Mountain Hydrology, Swiss Federal Institute for Forest, Snow, and Avalanche Research, Birmensdorf, 8093, Switzerland
[5]IDYST, Université de Lausanne, Lausanne, 1015, Switzerland
[†]deceased, 28 September 2022

*Correspondence to*: Marin Kneib (marin.kneib@gmail.com)

**Abstract.** Local snow redistribution processes such as avalanches can considerably impact the spatial variability of accumulation on glaciers. However, this spatial variability is particularly difficult to quantify with traditional surface mass balance measurements or geodetic observations. Here, we leverage high quality surface velocity and elevation change maps for the period 2012-2021 from Pléiades stereo images, and ice thickness measurements of Argentière Glacier (France) to invert for its distributed surface mass balance. This inversion is conducted using three different ice thickness modelling approaches constrained by observations, which all show a very good agreement between inverted surface mass balance and *in situ* measurements (RMSE < 0.96 m w.e. yr$^{-1}$ for the 11-year average). The detected spatial variability in surface mass balance is consistent between modelling approaches and much higher than what is predicted from an enhanced temperature-index model calibrated with measurements from a dense network of stakes. In particular, we find high accumulation rates at the base of steep headwalls on the left-hand side of the glacier, likely related to avalanching at these locations. We calculate distributed precipitation correction factors to reconcile the outputs from the enhanced temperature-index model with the inverted surface mass balance data. These correction factors agree with the outputs of a parametrization of snow redistribution by avalanching, indicating an additional 60% mass input relative to the accumulation from solid precipitation at these specific locations. Using these correction factors in a forward modelling exercise, we show that explicitly accounting for avalanches leads to twice more ice being conserved in the Argentière catchment by 2100 in an RCP 4.5 climate scenario, and to a considerably different ice thickness distribution. Our results highlight the need to better account for such spatially variable accumulation processes in glacio-hydrological models.



## 1 Introduction

Glacier surface mass balance (SMB) is traditionally measured using changes of the emerging length of stakes drilled into the ice in the ablation zone, and combined firn depth and density measurements in the accumulation (Cogley et al., 2011). These measurements, constituting the glaciological method, describe on an annual basis the spatio-temporal variability of SMB,
and enable linking local SMB and energy-balance with climate variables (Cuffey and Paterson, 2010). Glacier-wide SMB can be estimated from the interpolation of the distributed measurements. The glacier-wide SMB differs from geodetic mass balance, which is obtained by integrating the elevation change signal over the entire glacier using digital elevation models (DEMs) with a temporal baseline of a year or more, assuming a given volume to mass conversion factor (Berthier et al., 2023; Cogley et al., 2011; Huss, 2013). There can be a mismatch between the glacier-wide integration of local SMB
measurements using SMB gradients or models, and geodetic mass balance estimates (Cox and March, 2004; Huss et al., 2021; Rounce et al., 2020; Wagnon et al., 2021). This mismatch is due to internal processes, to density assumptions and to the spatial variability of SMB, controlled by local processes affecting the ablation (supraglacial debris of varying thickness, cliffs and ponds, topographic shading, calving) as well as the accumulation (wind redistribution or avalanching; e.g., Brun et al., 2018; DeBeer and Sharp, 2009; Jourdain et al., 2023; Réveillet et al., 2021; Voordendag et al., 2024; Zhao et al., 2023),
and that cannot be represented using local SMB measurements only.

In particular, avalanching leads to locally higher accumulation rates through the redistribution of snow from surrounding mountain headwalls onto the glacier surface (Benn and Lehmkuhl, 2000). These additional mass inputs have seldom been directly quantified using *in situ* measurements due to the difficulty and danger of accessing the avalanche deposits and the
high spatial variability of the accumulation patterns (Hynek et al., 2023; Mott et al., 2019; Purdie et al., 2015). While there is observational evidence from remote sensing that a large number of glaciers in the European Alps and High Mountain Asia are strongly avalanche fed (Kneib et al., 2023), very few studies have tried to quantify this contribution at the glacier scale. A limited number of studies have undertaken direct measurements of this accumulation (Hynek et al., 2023; Mott et al., 2019; Purdie et al., 2015), while others have relied either on the calibration of a 1D flowline model to find the missing
accumulation term explaining the higher than expected ice flux (Laha et al., 2017), or on parametrizations of mass redistribution from avalanching (Bernhardt and Schulz, 2010; Burger et al., 2018; Buri et al., 2023; Gruber, 2007; Mimeau et al., 2019). In other regional-scale models that rely on degree-day parametrizations to calculate the SMB, it is common to use a precipitation correction factor to account for this spatial variability in SMB, either for the entire glacier (Maussion et al., 2019; Rounce et al., 2020; Schuster et al., 2023) or locally at the base of headwalls (Gilbert et al., 2023; Rabatel et al., 2018).
For example, when deriving the ice thickness of Argentière Glacier using the Elmer/Ice model, Gilbert et al. (2023) had to apply a precipitation correction factor of 1.4 to their distributed surface mass balance estimates at the base of the headwalls in the upper accumulation zone of Argentière Glacier to be able to fit the observed ice flux.



Elevation change differences can indicate spatially variable signals caused by avalanching (Beraud et al., 2022; Pelto et al., 2019), but the quantification of this variability from elevation change requires accounting for the ice flux divergence (Jourdain et al., 2023; Vincent et al., 2021; Zeller et al., 2023). There have been a number of recent advances leveraging high quality velocity and thickness products to quantify the flux divergence, and therefore the distributed SMB of mountain glaciers. Initial efforts relied on flux gates to estimate and correct for emergence velocity in the ablation zone of debris-covered glaciers (Brun et al., 2018; Buri et al., 2021; Kneib et al., 2022; Miles et al., 2018; Mishra et al., 2021; Westoby et al., 2020; Zhao et al., 2023) and debris-free glaciers (Berthier and Vincent, 2012). The calculated emergence showed good agreement with *in situ* measurements at stakes (Berthier and Vincent, 2012), and this approach was applied to entire glaciers using distributed elevation change, ice thickness and surface velocity products available at the regional scale (Bisset et al., 2020; Cook et al., 2023; Miles et al., 2021; Pelto and Menounos, 2021). Fully distributed estimates of flux divergence can also be computed, but the spatial differentiation of the ice flux leads to numerical noises that need to be smoothed, at the cost of reduced accuracy and/or mass conservations issues, either using filters of variable lengths (Van Tricht et al., 2021), or by spatially aggregating the signal (Bisset et al., 2020; Miles et al., 2021; Pelto and Menounos, 2021). Such distributed SMB products have been used to quantify the melt rates of supraglacial ice cliffs and ponds on debris-covered glaciers (Brun et al., 2018; Kneib et al., 2022; Miles et al., 2018; Mishra et al., 2021; Westoby et al., 2020; Zhao et al., 2023), to estimate equilibrium-line altitudes and accumulation-area ratios (Miles et al., 2021), to invert for distributed debris thickness (McCarthy et al., 2022; Rounce et al., 2018) or to validate modelled SMB patterns (Buri et al., 2023). However, these estimates depend on the quality of the ice thickness, velocity and elevation change data which leads to high uncertainties particularly in the accumulation area of the glaciers (Miles et al., 2021).

In this study, we aim to: 1) produce high-resolution distributed SMB estimates for Argentière Glacier (French Alps) using different ice thickness modelling approaches and flux divergence calculation approaches; 2) evaluate these distributed SMB products against *in situ* measurements; 3) use these products to quantify the spatial variability of the SMB caused by local processes with a focus on avalanching; and 4) test the sensitivity of the glacier evolution to the spatial variability of accumulation.

## 2 Data and Methods

### 2.1 Site description

Argentière Glacier (45°55' N, 7°00' E) is located in the Mont-Blanc massif, European Alps (Fig. 1). The glacier extended from ~3500 m a.s.l. to ~1600 m a.s.l. at the terminus in 2022. Its surface area is ~12 km$^2$ for a length of ~10 km. It is surrounded by steep headwalls, especially on its left-hand side that release large avalanches onto the glacier surface (Fig. 1; Kneib et al., 2023). This glacier is particularly well studied and has numerous *in situ* measurements. The glacier surface mass balance has been continuously monitored since 1975 using a network of stakes (Fig. 1; Vincent et al., 2009) as part of





the GLACIOCLIM monitoring program (https://glacioclim.osug.fr/). There are also very high-resolution Pléiades digital elevation models (DEMs, 4 m resolution) and orthoimages (0.5 m resolution) available at a high temporal frequency (on average at least 2 per year) and since 2012 (Beraud et al., 2022). In addition, there is a relatively high density of ice thickness measurements from ground penetrating radar (GPR; Fig. 1; Rabatel et al., 2018).

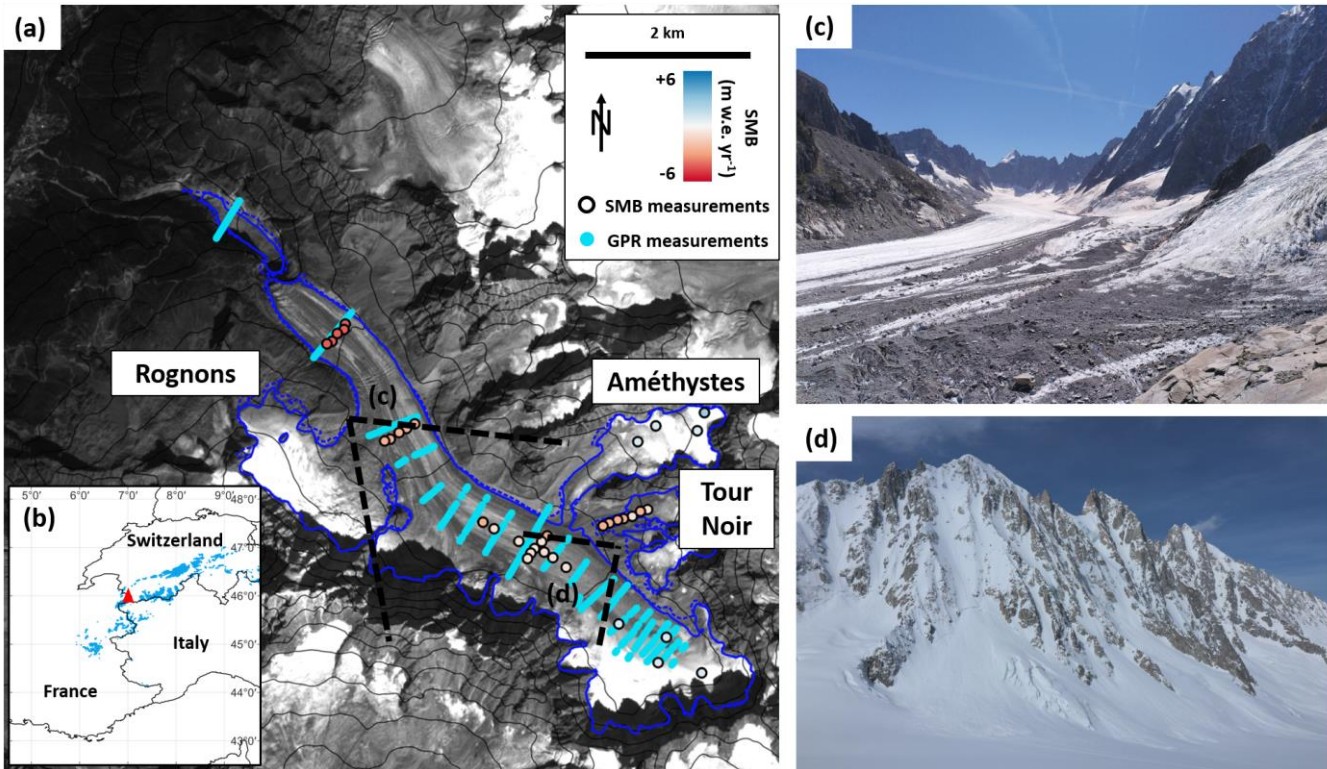


**Figure 1: (a) Map of Argentière Glacier (blue outlines) with ground penetrating radar (GPR) transects in turquoise and stake locations in circles with the average annual mass balance for the period 2012-2021 indicated by the colour scale. These values range between -5.3 and +2.3 m w.e. yr⁻¹. The blue glacier outlines were derived from a Pléiades orthoimage acquired on 08/09/2020 and the dashed blue glacier outlines were derived from a Pléiades orthoimage acquired on 19/08/2012. The black elevation contour lines are spaced every 200 m. The black dashed lines indicate the viewing angle of the pictures shown in (c) and (d) taken on 11/08/2023 and 05/05/2023, respectively. Background image is a Sentinel-2 scene (band 04) from 11/09/2022. (b) Overview map with Argentière Glacier indicated by the red triangle. Randolph Glacier Inventory (RGI) 6.0 outlines shown in blue.**

## 2.2 Glacier outlines

We manually derived glacier outlines based on the 0.5 m resolution Pléiades 09/08/2020 and 19/08/2012 orthoimages. For the 2020 outlines we also used the Pléiades velocity data (section 2.4) to remove stagnant zones with a velocity lower than 1 m yr⁻¹, which we considered to not contribute any ice flux to the rest of the glacier. In particular, this led to the removal of ice bodies above the highest bergschrunds (Fig. 1a; Nuimura et al., 2015). On the right-hand side of Argentière Glacier, the Glacier du Tour Noir (flowing west) is disconnected from the main glacier trunk and the Glacier des Améthystes is only connected via a thin tongue of ice (Fig. 1). On the left-hand side, flowing towards the north-east, the Glacier des Rognons is





still well connected despite a narrowing connection. At these locations especially, the glacier outlines have changed considerably since 2012 (Fig. 1).

## 2.3 Elevation change

We used 13 DEMs processed at 4-m resolution from Pléiades stereo-images acquired between 19/08/2012 and 15/08/2021, with at least one DEM per year, except for 2014. These DEMs were all acquired in the end-of-summer months, between the
12th of August and the 30th of September to reflect surface elevation at the end of the melt period and reduce uncertainties in the coregistration by limiting the presence of snow on the surrounding stable terrain (Beraud et al., 2022). These DEMs were all co-registered following the workflow developed by Beraud et al. (2022). We then interpolated an elevation change trend for all pixels of the stacked DEMs with at least eight observations over at least five years using a linear regression (Berthier et al., 2016). We filtered out unrealistic values below -10 m yr$^{-1}$ or above 5 m yr$^{-1}$. Remaining gaps (2.2% of the
glacier area) were filled using a spatial cubic interpolation and the resulting trend was spatially smoothed with a 3x3 median filter. Uncertainties were calculated based on the surrounding off-glacier stable terrain. We also computed a reference mean DEM in the middle of the study period (15/02/2017) from the DEM with the least data gaps (from 09/08/2020), to which we applied the temporal trend.

## 2.4 Surface velocity

We computed the velocity of Argentière Glacier using 277 pairs of Pléiades 0.5-m resolution orthoimages acquired over the Mont-Blanc massif and covering the period 08/08/2012-13/02/2022. The velocity fields were obtained using normalised cross-correlation and the images were co-registered using the median velocity of the off-glacier terrain (Millan et al., 2019, 2022). For each pixel of the glacier we removed the velocities that deviated from the mean flow direction over the period by more than 15°. We did not remove the pairs with a high standard deviation on the off-glacier terrain, as done in previous
studies (Dehecq et al., 2019). The noise in these pairs arises from the lack of features to correlate when the off-glacier terrain is covered with snow, but the signal on the glacier is still consistent, especially in crevassed-areas. We computed the median and standard deviation from this stack of 2D velocity fields, and estimated the uncertainty, $\sigma_{u_s}$, as the sum of the mean and standard deviation on the off-glacier stable terrain. The median velocity field was spatially smoothed using an 11x11 px median filter. The remaining gaps (14.7% of the glacier area) were filled bilinearly.

## 2.5 Ice thickness


We used distributed ice thicknesses obtained from three different modelling approaches constrained by *in situ* ice thickness observations. We used the GPR measurements of the glacier bed presented in the study by Rabatel et al. (2018) and adjusted the ice thickness to our 2017 reference DEM. This data consists of 21 cross-sectional transects along the main glacier flowline (Fig. 1a). The three ice thickness modelling approaches that we used are:



- **The SIA thickness**, which we derived using the Shallow Ice Approximation (SIA) that allows to express the ice thickness $H$ as a function of surface velocity ($u_s$) and slope ($s$; Millan et al., 2022):

$$H = \left(\frac{u_s(1-\beta)(n+1)}{2A(\rho g)^n \|\nabla s\|^n}\right)^{\frac{1}{n+1}},$$ (1)

  Where $\beta$ is the ratio between basal and surface velocity, $n$ is the exponent in Glen's flow law which we assume to be equal to 3, $A$ is the creep parameter, $\rho$ is the ice density taken as 917 kg m$^{-3}$ and $g$ is the gravitational acceleration. We calibrated $A$ and $\beta$ at the locations of the GPR measurements by minimising the root mean square error (RMSE) between the model and the observations, and used this equation to extrapolate the thickness to the entire glacier.

- **The F2019 thickness** estimate from the global product by Farinotti et al., (2019) which was originally derived from five different estimates of various sources. This is a reference product that is available for all mountain glaciers in the world, and which did not use the GPR measurements made on Argentière.

- **The IGM thickness,** which was obtained using the inversion capability of the Instructed Glacier Model (IGM) constrained by the ice thickness profiles, surface velocity, and glacier surface elevation data (Jouvet, 2023; Jouvet and Cordonnier, 2023). IGM emulates the 3D Blatter-Pattyn ice flow model using convolutional neural networks (Jouvet, 2023; Jouvet and Cordonnier, 2023). Starting from a given initial thickness (here, the SIA thickness), IGM iteratively updates it to reach an optimised solution after a given number of iterations (1000 in our case), using weights on the different control variables. The final outputs include optimised ice thicknesses and corresponding 3D velocity fields which are used to directly compute the flux divergence using centred finite differences. The flux divergence is smoothed by enforcing a linear regression with the glacier surface elevation in the optimization process (Jouvet and Cordonnier, 2023). This optimised flux divergence can then be used directly to compute the distributed surface mass balance (section 2.6). As with the other approaches, the flux divergence calculation does not exactly conserve mass (section 2.6), and the resulting mass excess or shortage is then redistributed homogeneously to the entire glacier.

The SIA thickness inversion was run at 20 m resolution and the F2019 thickness was bilinearly resampled from 25 to 20 m. For these first two modelling approaches, uncertainties in the distributed ice thickness were determined using sequential gaussian simulations (SGS, 100 for each modelling approach) based on variograms of the residuals between the modelled beds and the observations. These simulations enable varying the ice thickness within uncertainty bounds while preserving the spatial smoothness of the observations. As such, they add more uncertainty where no observations are available, at locations away from the glacier outlines and GPR measurements (Goovaerts, 1997; MacKie et al., 2021). Due to their high



computational cost, the SGSs were run at 50 m resolution and their outputs were then bilinearly resampled to 20 m for the SMB inversion.

## 2.6 Surface mass balance inversion

The main objective of this study is to compute the distributed surface mass balance at 20 m resolution for the entire
Argentière Glacier using mass conservation considerations. From a Eulerian perspective, one can write the mass conservation equation at each point of the glacier surface (Cuffey and Paterson, 2010; Hubbard et al., 2000; Miles et al., 2021):

$$\frac{\rho_{dH}}{\rho_{H_2O}}\frac{dH}{dt} = \dot{b} - \frac{\rho_{\nabla q}}{\rho_{H_2O}}\nabla * \boldsymbol{q}, \tag{2}$$

Where $\nabla * \boldsymbol{q}$ is the flux divergence (in kg m$^{-2}$), $\frac{dH}{dt}$ is the rate of elevation change, and $\rho$ is the density of each term, with $H_2O$
standing for liquid water. Here we assume that the basal and internal mass balance are negligible and, in the accumulation area, that firn densification rates do not change over time; $\dot{b}$ is therefore equal to the surface mass balance (in m w.e. yr$^{-1}$). For all that follows, upward direction is indicated with a positive sign and downward direction with a negative sign. Following the shallow ice approximation, the flux $\boldsymbol{q}$ can be expressed in each location as a function of ice thickness $H$ and surface velocity $\boldsymbol{u_s}$:

$$\boldsymbol{q} = h\gamma\boldsymbol{u_s}, \tag{3}$$

Where $\gamma\boldsymbol{u_s}$ represents the column-averaged velocity (Miles et al., 2021). This approximation allows for the fast computation of the flux, without solving the full 3D Stokes equations for the SIA and F2019 modelling approaches, but this implies making an assumption on the sliding regime of the studied glacier (Hubbard et al., 2000). This step was however not necessary for the IGM modelling approach, which directly outputs the flux divergence (section 2.5).


Computing the flux divergence directly from the thickness and surface velocity results in a highly noisy and unrealistic signal (Van Tricht et al., 2021) caused by uncertainties and noise in local thickness and velocity data that are enhanced by the divergence calculation and because the flux divergence depends not only on the local geometry but, due to longitudinal stress, on the surrounding geometry over scales of several ice thicknesses (Kamb and Echelmeyer, 1986; Zekollari et al.,
2014). For the SIA and F2019 modelling approaches we therefore smooth the flux divergence using a local gaussian filter, with a scaling length equal to four ice thicknesses (Le Brocq et al., 2006; Van Tricht et al., 2021). This filter is not fully mass conservative, due to discontinuities at the glacier edges, and the mass excess or shortage resulting from this smoothing is then redistributed homogeneously to the entire glacier.

To link local volumetric changes to mass changes, the relevant densities (Eq. 2) must be determined for all three modelling approaches. In our implementation, the density of the flux is set at 900 kg m$^{-3}$ for the entire glacier. The density of the



elevation change signal in the ablation area (where flux divergence is positive and the elevation change is negative) is set to 900 kg m$^{-3}$ and 600 kg m$^{-3}$ in the accumulation area (where flux divergence is negative and elevation change positive; Miles et al., 2021; Table S1). At locations where flux divergence and elevation change have different signs, the density of the

elevation change signal should be between 600 and 900 kg m$^{-3}$, and we assume a uniform distribution of values within this range. The $\gamma$ value to convert the surface to the column averaged velocity for the entire glacier is assumed to be within [0.8; 1], with a uniform distribution of values (Table S1). The extreme values of 0.8 and 1 correspond respectively to a shearing-dominated flow and to a sliding-dominated flow (Cuffey and Paterson, 2010).

**2.7 Choice of parameters and uncertainty propagation**

We computed the flux divergence following three different modelling approaches, leading to three SMB estimates (Eq. 2), the **SIA, F2019** and **IGM** estimates. The flux and SMB calculations from the F2019 and SIA thicknesses were all conducted using inputs resampled to 20 m resolution, bilinearly for the thickness, velocity and flux divergence, and using a natural neighbour interpolation for the elevation change. The IGM inversion was run at 50 m, and the resulting flux divergence fields were then bilinearly resampled to 20 m to obtain the distributed SMB.


We propagated uncertainties using a Monte Carlo approach with 1000 runs for the SIA and F2019 modelling approaches, by perturbing the $\gamma$ value, the density of the flux in the zone where both flux divergence and elevation change have the same sign, the velocity, the elevation change and the ice thickness signals. We assumed a uniform distribution of density of the elevation change signal in the mixed zone between 600 and 900 kg.m$^3$, and of $\gamma$ between 0.8 and 1. For the ice thickness, we

used for each thickness estimate the set of 100 sequential gaussian simulations to randomly draw from (section 2.5). For all the other parameters we assumed a normal distribution of uncertainty. We also conducted a sensitivity analysis for the SIA and F2019 modelling approaches following a one-at-time sensitivity test by running the SMB inversion 100 times for each individual parameter, while keeping the others fixed.

For the IGM estimate, we computed the flux divergence from the IGM inversion 100 times varying the weights on the surface elevation, velocity and thickness observations uniformly between [0 0.5] m, [0 $\sigma_{u_s}$] m yr$^{-1}$ and [0 50] m, respectively (Table S1). These weights correspond to the uncertainty of the different constraints, and can therefore be interpreted as a tolerance of misfit to these variables. We then applied a Monte Carlo approach with 1000 runs, by randomly drawing from the 100 flux divergence fields, and perturbing the density of elevation change in the mixed zone and the elevation change

using the same distributions as for the F2019 and SIA modelling approaches.



## 2.8 *In situ* SMB observations

We used annual surface mass balance measurements taken at stake locations over the period 2012-2021. These measurements are conducted every year by the GLACIOCLIM monitoring programme (https://glacioclim.osug.fr/) in the accumulation and ablation areas of Argentière, Tour Noir and Améthystes glaciers (Fig. 1a). We averaged the annual surface
mass balances at the location of each stake over the period 2012-2021, which highlighted a different altitudinal pattern for the Argentière and Tour Noir/Améthystes stakes (Fig. 2a). In all that follows, we refer for simplicity to the Tour Noir and Améthystes stakes as the Tour Noir stakes. We compared the inverted distributed SMB with the *in situ* SMB at the GLACIOCLIM stake locations and for each modelling approach selected the best 10% SMB estimates that minimised the weighted quadratic sum of the RMSE of the Argentière and Tour Noir stakes.

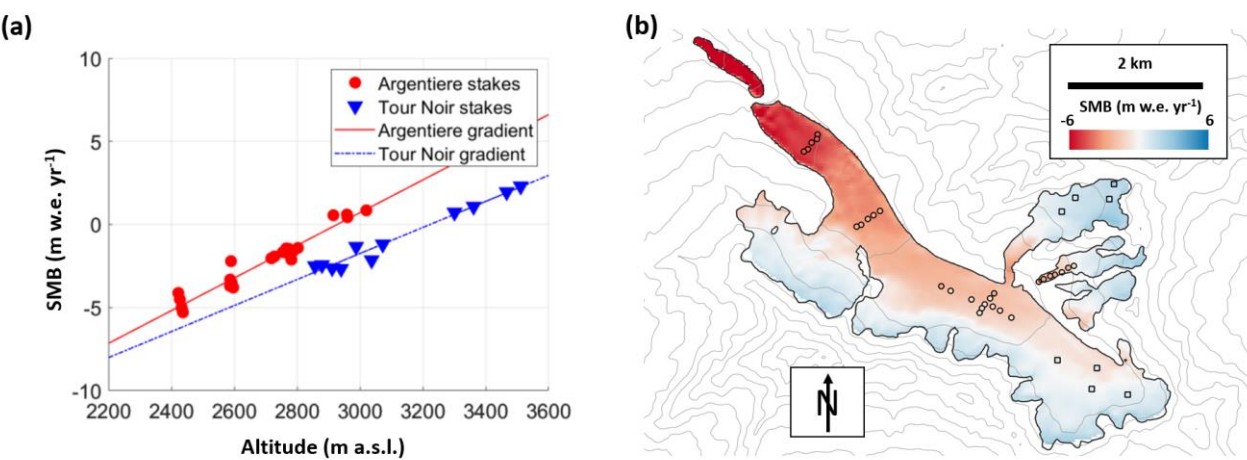


**Figure 2: (a) average annual SMB for the period 2012-2021 at the stake locations on Argentière Glaciers (red dots) and Tour Noir and Améthystes Glaciers (blue triangles), and corresponding altitudinal SMB gradients. (b) Mean annual 2012-2021 stake measurements at the stake locations (circles indicate stakes in the ablation zone and squares show stakes in the accumulation zone), and mean annual SMB over the period 2012-2021, as obtained from the GLACIOCLIM ETI SMB model (section 2.10). The**
**grey elevation contour lines are spaced every 200 m. The black glacier outlines were derived from a Pléiades orthoimage acquired on 08/09/2020.**

For the validation of the spatial patterns of the inverted SMB, we manually extracted end-of-summer snow lines from all available Pléiades orthoimages that were unaffected by fresh snowfall between August 14th and September 17th over the study period. This was possible for the years 2012, 2015, 2017, 2018, 2019, 2020 and 2021.


We also conducted GPR snow thickness surveys with a MALÅ ProEx control unit equipped with a 250 MHz shielded antenna using a trigger interval of 1 s, a 256-fold stacking, and a time window of 105 ns, which was the same system used by Jourdain et al. (2023). These GPR surveys were conducted on 25/05/2018, 06/05/2019 and 28/04/2022 at the end of the accumulation season, and followed similar paths, with the 2018 survey being the shortest (Fig. S1). The wave propagation
velocity in the snow to estimate the snow depth above the previous year's horizon was calibrated with manual snow depth



measurements conducted on the same day as the surveys as part of the GLACIOCLIM SMB measurements (Jourdain et al., 2023). We used these measurements as indicators of the variability in the snow accumulation, and averaged them along the 2018 survey path, using cubic interpolation when the tracks did not coincide perfectly (Fig. S1). We converted the mean snow heights to snow water equivalent using a density of 440 kg m$^{-3}$, which was the mean density measured at all snow

profiles in 2018, 2019 and 2022 in the accumulation area of Argentière Glacier.

## 2.9 Reference SMB model

We used a SMB model derived from the enhanced temperature index (ETI) model used in Gilbert et al. (2023). In all that follows this model will be referred to as the GLACIOCLIM ETI model, run at daily time-step:

$$SMB = A - M, \tag{4}$$

where $A$ is the local daily snow accumulation (m w.e. d$^{-1}$) and $M$ the local daily surface melt (m w.e. d$^{-1}$). The amount of melt $M$ is computed from the available energy for melt $Q_m$ following Oerlemans, (2001):

$$Q_m = (1 - \alpha)rI_{pot} + kT + k_0, \tag{5}$$

Where $\alpha$ is the local surface albedo (= $\alpha_{ice}$ or = $\alpha_{snow}$, which decreases exponentially with the age of the surface snow), $I_{pot}$ is the potential incoming shortwave radiation, $r$ is a corrective factor, $T$ is the air temperature (°C) and $kT + k_0$ is a

parametrization of the longwave radiation balance and the turbulent heat exchange linearized around the melting point with $k$ depending on the surface state (= $k_{ice}$ or = $k_{snow}$) (Réveillet et al., 2017) and $k_0$ a constant. If $Q_m$ is greater than zero, the melt rate is obtained with the latent heat of fusion. The evolution of snow and firn thicknesses is computed at a daily time step to determine the albedo value. The local snow accumulation $A$ is determined from local daily precipitation $P$ according to a snow/rain temperature threshold fixed at 1°C ($A = P$ if $T<1$°C and $A = 0$ otherwise).


Temperature and precipitation data are provided at one elevation and distributed to the glacier surface according to altitudinal lapse rates such as:

$$T(z,t) = T_{ref}(t) + \frac{dT}{dz}(z - z_{ref}), \tag{6}$$

$$P(x,y,z,t) = k_{P_{ref}} * P_{ref}(t)(1 + \frac{dP}{dz}(z - z_{ref})), \tag{7}$$

where $z$ is the elevation of the surface (m a.s.l.), $T_{ref}(t)$ and $P_{ref}(t)$ are air temperature and precipitation time series at the elevation $z_{ref}$, $dT/dz$ is the temperature lapse rate (°C m$^{-1}$), $k_{P_{ref}}$ is a correction factor for the precipitation at the elevation $z_{ref}$ and $dP/dz$ is the precipitation lapse rate (% m$^{-1}$).

The GLACIOCLIM ETI model was run with S2M temperature and precipitation data at 2400 m ($z_{ref}$; Vernay et al., 2022)
and the different parameters were calibrated against the GLACIOCLIM measurements and geodetic mass balance measurements from the period 1975-2020 (Gilbert et al., 2023). The precipitation lapse rate $dP/dz$ was 0.05 % m$^{-1}$ and the





correction factor $k_{P_{ref}}$ was 1.2. The distributed model was run at 40 m resolution bilinearly resampled to 20 m, and the resulting annual surface mass balances were averaged over the study period (Fig. 2b). In the original study by Gilbert et al. (2023) there was also an arbitrary additional precipitation correction factor ($P_{fact}$) that was imposed at the base of the

headwalls on the left side of Argentière Glacier to account for avalanching. Here, we re-evaluated this $P_{fact}$ over the Rognons, Tour Noir and Améthystes tributaries and in the accumulation area using the mean of the best 10% inverted SMB patterns of each modelling approach ($SMB_{inverted}$) and the mean yearly distributed accumulation and melt from the GLACIOCLIM ETI SMB model over the period 01/08/2012-01/08/2021:

$$SMB_{inverted} = A * P_{fact} - M, \tag{8}$$

As these anomalies cannot solely be attributed to additional accumulation, and as there are feedbacks caused by adding or removing snow in the model, we iterated this step until the modelled SMB with the additional $P_{fact}$ converged to a stable value for the period 2012-2021. These precipitation correction factors were then used to modify the surface mass balance (corrected ETI model) for the forward modelling of Argentière Glacier (section 2.11).

## 2.10 Zones influenced by avalanches

We could identify the potential areas of avalanche contribution using two independent approaches:

- We identified the avalanche locations mapped from Sentinel-1 images over the 01/11/2016-31/10/2021 period from the dataset by Kneib et al. (2023).
- We used a snow redistribution model based on a parametrization of the maximum snow height for a given slope to redistribute the excess snow using a multiple flow direction routing scheme (Bernhardt and Schulz, 2010; Ragettli
et al., 2015). This model was run with the lapsed solid precipitation data from the GLACIOCLIM ETI model (section 2.10), using the original parameters from Bernhardt and Schulz (2010), for the period 01/08/2012 to 01/08/2021 at 60 m resolution and a monthly time-step. The resulting monthly snow accumulations were then summed for each year and these annual accumulation rates were averaged over the entire study period.

These approaches were used to qualitatively (first approach) and quantitatively (second approach) compare the precipitation
correction factors ($P_{fact}$) with the potential mass inputs from avalanches.

## 2.11 Forward modelling

In order to test the influence of avalanching on the future simulations of Argentière Glacier, we ran full Stokes simulations adopting Glen's flow law with $n = 3$ for viscous isotropic temperate ice (Cuffey and Patterson, 2010) with the finite element model Elmer/Ice (Gagliardini et al., 2013) for the periods 1907-2023 (historical runs) and 2023-2100 using RCP 4.5 CMIP 5
climate data (future runs) from the Euro-Cordex project downscaled following the ADAMONT approach (Verfaillie et al., 2017). More details are provided in Gilbert et al. (2023), where the authors used the same model implementation and



calibration with long-term geodetic data and *in situ* measurements. The objective here is not to provide detailed scenarios of the evolution of Argentière Glacier but rather to test the sensitivity of the glacier to the spatial variability of the SMB.

We used two SMB scenarios for the forward runs:

- The ***GLACIOCLIM*** scenario used the GLACIOCLIM ETI model (without $P_{fact}$) calibrated against the stake measurements over the period 1975-2020, and glacier-wide geodetic mass balance observations for the years 1904, 1949, 1970, 1980, 1994, 1998, 2003, 2008 and 2019. Such calibration strategy is what is most commonly done when both *in situ* and geodetic observations are available, but without any other information on the spatial
variability of the SMB.

- The ***Corrected*** scenario which uses the corrected ETI model, *i.e.* the GLACIOCLIM ETI with additional $P_{fact}$ calibrated against the inverted SMB scenarios over Rognons, Tour Noir and Améthystes tributaries and in the accumulation area (section 2.9). This scenario therefore also integrates the geodetic information, but the spatial variability of the SMB is supposed to be closer to reality.

**3 Results**

**3.1 Distributed $dH/dt$, surface velocity and ice thickness**

Over the period 2012-2021, the surface elevation change across Argentière Glacier ranges between -6 m yr$^{-1}$ in the small zone at the terminus that is disconnected from the main glacier trunk, and +2 m yr$^{-1}$ in some local areas at the base of the upper headwalls (Fig. 3a). The average value over the whole glacier is equal to -0.85 m yr$^{-1}$. The mean elevation change
value over the stable terrain is -0.002 m yr$^{-1}$ and the standard deviation, which we consider to be the 1-σ uncertainty of $dH/dt$, is equal to 0.07 m yr$^{-1}$.

The median surface velocity ranges between 0 and 150 m yr$^{-1}$. This maximum velocity is reached on the densely crevassed Rognons Glacier (tributary on the left-hand side) but for the rest of Argentière Glacier the mean velocity remains lower than
80 m yr$^{-1}$ (Fig. 3b). Both the mean and standard deviation of velocity on the off-glacier terrain are equal to 1.2 m yr$^{-1}$, leading to an uncertainty in velocity of 2.4 m yr$^{-1}$.



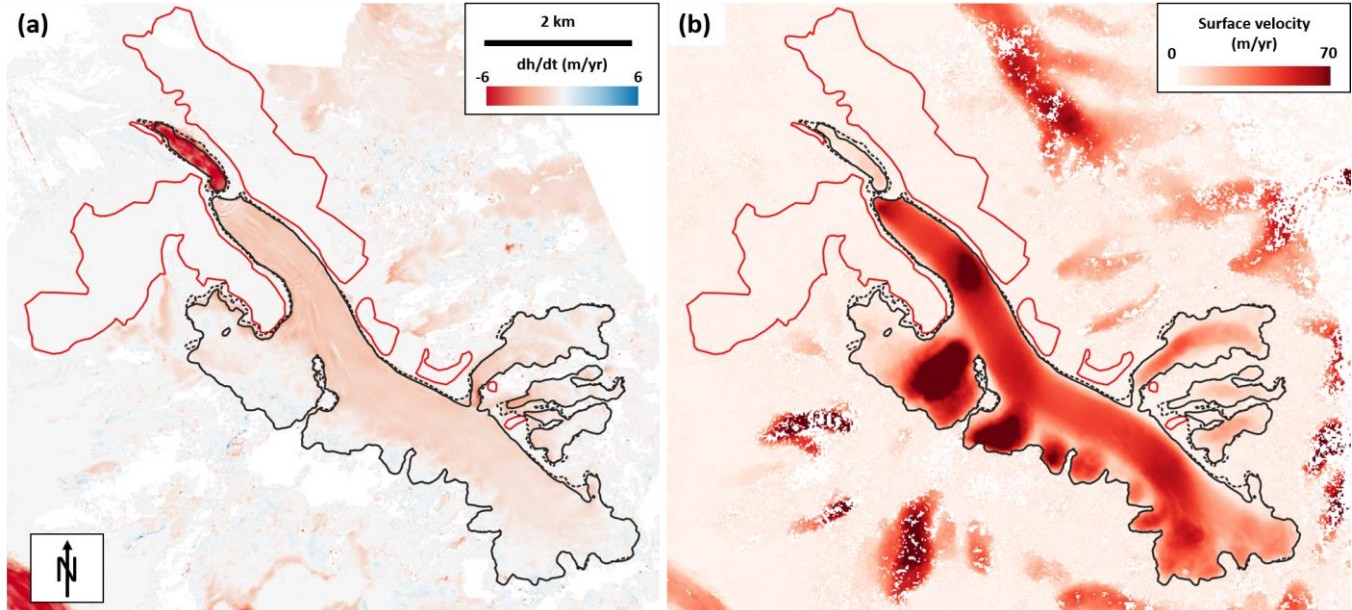

**Figure 3: (a) Distributed elevation change rate over the 2012-2021 period from the 13 end-of-summer Pléiades DEMs. (b) Mean velocity over the 2012-2022 period from all 277 Pléiades orthoimage pairs. The red outlines show the zones of off-glacier stable terrain used to estimate the uncertainties of these products. The black outlines indicate the glacier outlines manually derived from the 08/09/2020 Pléiades orthoimage, and the dashed outlines show the glacier outlines derived from the 19/08/2012 Pléiades orthoimage.**

*In situ* GPR measurements indicate thicknesses of up to 464 m along the main trunk of Argentière Glacier for the reference mean DEM of 15/02/2017 (Fig. 4a). All three modelled ice thickness products show similar patterns, with the ice being thickest along the main glacier trunk, with more or less overdeepenings (Fig. 4b-d). The F2019 thickness estimate is the shallowest, with a maximum thickness of 266 m (Fig. 4b). The mean thickness from the 100 IGM inversions has a maximum thickness of 354 m but a narrower shape, which results in a similar volume than the F2019 modelling approach (Fig. 4d).





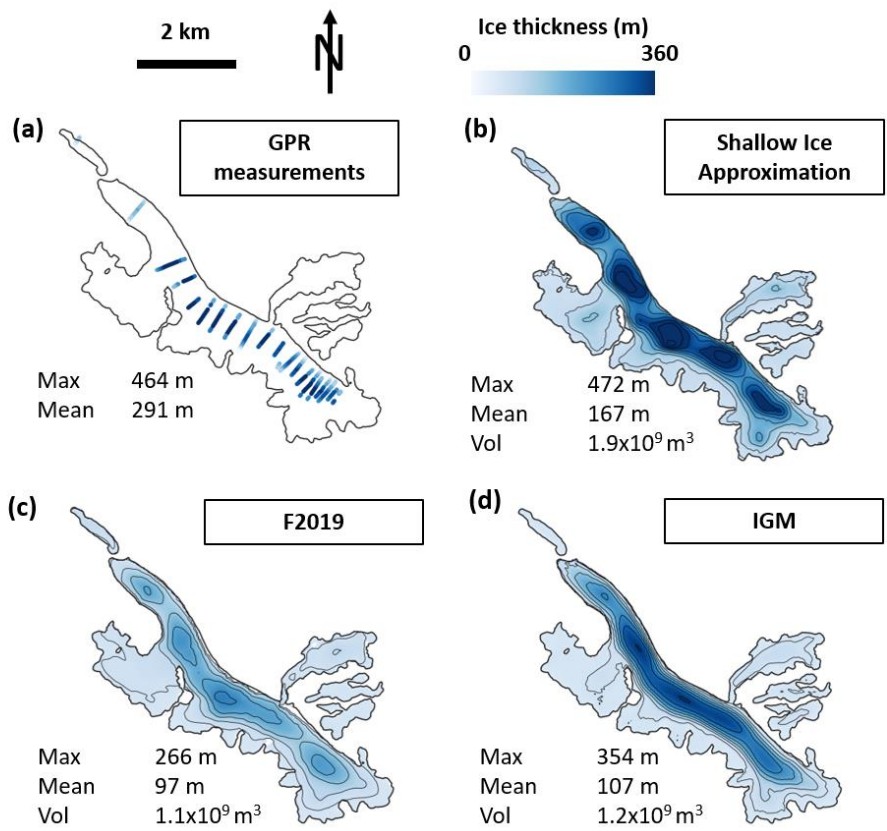

**Figure 4: (a) Ice thickness from in situ GPR measurements. (b) Distributed ice thickness obtained using the shallow ice approximation. (c) Distributed ice thickness from the Farinotti et al. (2019) estimate. (d) Mean distributed ice thickness from the 100 ice thickness inversions obtained with IGM. The black thickness contour lines are spaced every 50 m. The black glacier outlines were derived from the 08/09/2020 Pléiades orthoimage. The numbers in each panel indicate the maximum and mean thickness and the total glacier volume.**

The mean thicknesses of all SGSs result in similar patterns and values than the modelled thicknesses (Fig. S2-S3), but the standard deviation increases with the residuals between modelled and observed thicknesses, and in between observations (Fig. S2-S3). The F2019 thickness has the highest uncertainties (standard deviation up to 85 m) followed by the SIA thickness (up to 72 m). The IGM thicknesses mostly differ at the lateral margins of the main glacier trunk and on the tributaries with no GPR measurements, with a maximum standard deviation of 43 m (Fig. S4b).

## 3.2 Surface mass balance patterns

All three modelling approaches lead to similar distributed SMB patterns (Fig. 5-6) that agree with the end-of-season snowlines extracted from the Pléiades images (Fig. S5). At the same elevation, the spatial variability in SMB along the main glacier trunk remains low, below 2 m w.e. yr$^{-1}$ for all modelling approaches (Fig. 6a-c). The Rognons tributary has distinct



lower SMB values between 2600 and 2800 m a.s.l., leading to differences of up to 5 m w.e. yr$^{-1}$ compared to the main glacier trunk at the same elevation. These differences are more pronounced for the SIA modelling approach. Above 2700 m a.s.l.,
this altitudinal SMB variability on Argentière Glacier strongly increases for all modelling approaches, with spreads reaching up to 15 m w.e. yr$^{-1}$ between 3000 and 3100 m a.s.l. for the SIA model. The comparison with the measured GLACIOCLIM SMB indicates both high correlation coefficients ($R^2$>0.91) and low root mean square error (RMSE<0.96 m w.e. yr$^{-1}$). The SIA and F2019 inversions tend to slightly underestimate the SMB at the Tour Noir stake locations, particularly in the accumulation area. The SMB from the IGM inversion at these locations has the lowest RMSE (0.59 m w.e. yr$^{-1}$, Fig. 6f) but
also displays a higher spatial variability (Fig. 5c, 6c).

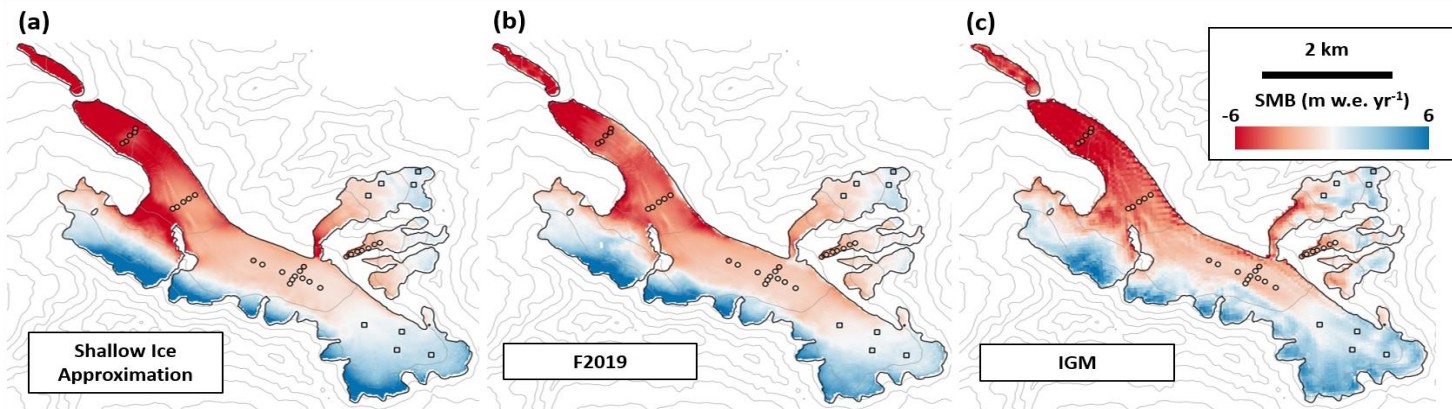

**Figure 5: SMB obtained with (a) the SIA, (b) the F2019 and (c) the IGM modelling approaches. The coloured dots indicate the mean annual mass balances over the period 2012-2021 from the stake measurements. The black glacier outlines were derived from a Pléiades orthoimage acquired on 08/09/2020.**

There is a good agreement with the SMB measurements on Argentière for both the SIA and F2019 thickness with RMSE values lower than 0.67 m w.e. yr$^{-1}$ (Fig. 6d-e), but with higher uncertainties obtained from the Monte Carlo simulations, up to +/- 1.2 m w.e. yr$^{-1}$ for the F2019 thickness modelling approach and +/- 0.8 m w.e. yr$^{-1}$ at the stake locations and higher at the margins of the glacier, particularly over the Rognons tributary (Fig. S6), where they locally reach up to +/- 6-8 m w.e. yr$^{-1}$. While not directly comparable as they were not obtained in the same way, these uncertainty patterns are similar for the
IGM modelling approach, reaching at most +/- 1.1 m w.e. yr$^{-1}$ at the stake locations and +/- 7 m w.e. yr$^{-1}$ on the Rognons tributary (Fig. S6). This is also where the 100 simulations of the IGM inversion differ the most in terms of surface velocity, with a standard deviation of 30 m yr$^{-1}$ while it is lower than 7 m yr$^{-1}$ elsewhere (Fig. S7c). In the upper accumulation area of Argentière Glacier there is also more spatial variability in SMB predicted by each of the three modelling approaches than by the GLACIOCLIM ETI model. This enhanced spatial variability shows an improved agreement with the Spring GPR snow
accumulation measurements (Fig. S8).





For the SIA and F2019 modelling approaches, the uncertainty in the ice thickness is the main driver of the overall SMB uncertainty, followed by the uncertainty in velocity, $\gamma$ ratio, elevation change and density of elevation change signal in the mixed zone (Fig. S9, S10). The spatially-averaged mean uncertainty resulting from the ice thickness uncertainties varies

between 1.1 and 1.6 m w.e. yr$^{-1}$, while the uncertainty resulting from the velocity uncertainty is lower than 0.3 m w.e. yr$^{-1}$ for all modelling approaches, and the one resulting from the $\boldsymbol{\gamma}$ ratio is lower than 0.13 m w.e. yr$^{-1}$ (Fig. S9). The uncertainty in the IGM SMB is driven by the ratios between the weights of the thickness, surface velocity and elevation observations (Fig. S11). The lowest RMSE values between the IGM SMB and the GLACIOCLIM SMB for Argentière are obtained for ratios of thickness *versus* velocity and elevation weights close to 100 (Fig. S11d-e). For the Tour Noir tributaries, which do not

have any thickness measurements, the lowest RMSE values are obtained when the weights on the surface velocities are lower than the weights on the surface elevation (Fig. S11f). The uncertainty on the elevation change or density of elevation change in the mixed zone only have a very minor effect on the RMSE for Argentière, but a more important one for the Tour Noir, with RMSE values decreasing with increasing elevation change bias (Fig. S11g-h).





**Figure 6: (a-c)** Altitudinal patterns of mean annual SMB calculated with the different modelling approaches and of the mean annual mass balances over the period 2012-2021 from the stake measurements. **(d-f)** Direct comparison of mean annual SMB calculated with the different modelling approaches, with the mean annual mass balances over the period 2012-2021 from the stake measurements, at the stake locations.



### 3.3 Deviations from the GLACIOCLIM ETI model

For all distributed SMB modelling approaches, we obtain on the left-hand side of Argentière Glacier higher values than expected using the GLACIOCLIM ETI model (Fig. S12a-b). These higher values generally coincide with avalanche deposits at the base of headwalls, as observed from remote sensing radar images (Fig. 7a) or using a simple snow redistribution model (Fig. 7b). We partitioned the survey domain into five different zones and computed the $P_{fact}$ that would reconcile the modelled and inverted SMB (Fig. 8). All three modelling approaches lead to consistent spatial patterns and indicate similar

precipitation factors. These are highest (between 1.6 +/- 0.5 and 1.7 +/- 1.1) on the left-hand side of the glacier at the base of the steepest and tallest headwalls. This is also where most modelled redistributed snow from the SnowSlide parametrization accumulates, leading to a $P_{fact}$ of 1.7. Upon iterating the ETI model to account for the retroactions of adding snow to the glacier surface, the $P_{fact}$ value still reaches 1.6 for this zone (Fig. S12d). All three modelling approaches also indicate similar factors, between 1.2 +/- 0.5 and 1.5 +/- 0.3, in the *Fond du Cirque* zone, slightly less than the SnowSlide parametrization

(1.6). After iteration, the mean $P_{fact}$ obtained for the corrected ETI model is also 1.3 in this zone. On the main glacier trunk, where most stakes are located, the predicted $P_{fact}$ are between 0.7 +/- 0.3 and 0.8 +/- 0.5, indicating a lower SMB than the GLACIOCLIM ETI model. All three modelling approaches consistently indicate very low SMB on the lower part of the Rognons tributary of Argentière Glacier, leading to $P_{fact}$ lower than 0 for the SIA and IGM modelling approaches in this particular location (Fig. 8). After iteration, the mean correction factor obtained is 0.7, highlighting the compensating

influence of retroactions within the model. The $P_{fact}$ are also lower than 1 (between 0.5 +/- 0.3 and 0.7 +/- 0.2) across the Améthyste and Tour Noir tributaries, contrary to what is predicted by the snow redistribution model (1.2). When aggregated over the entire Argentière Glacier (with the Améthyste and Tour Noir tributaries), the $P_{fact}$ varies between 0.9 +/- 0.4 (SIA) and 1.0 +/- 0.6 (F2019), with a final $P_{fact}$ of 1.1, indicating an overall compensation of the high SMB anomalies at the base of the headwalls and the low anomalies in the Améthyste and Tour Noir tributaries and the main glacier tongue.



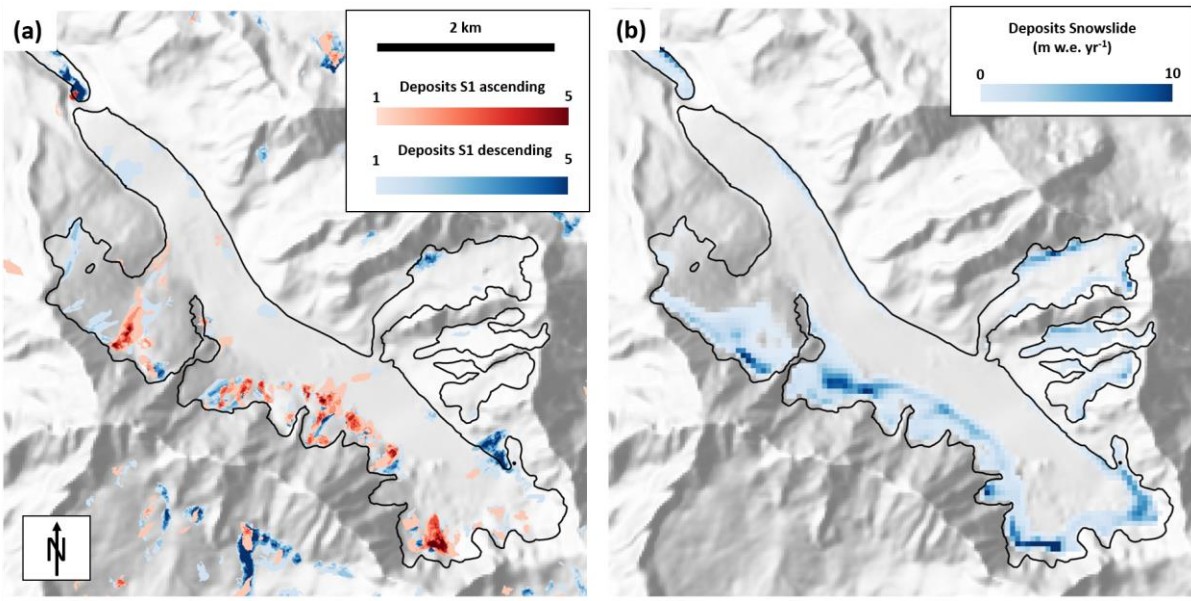


**Figure 7: (a) Total number of detected avalanches on Argentière Glacier for the period 11/2016-10/2021 using Sentinel-1 synthetic aperture radar images at 6 days intervals, both in the ascending (red) and descending (blue) orbits. Data from Kneib et al. (2023). (b) Average yearly snow accumulation by avalanches predicted by the SnowSlide model for the period 2012-2021. Background image is the hillshade of the AW3D30 30 m DEM (Tadono et al., 2014). The black glacier outlines were derived from the**
**08/09/2020 Pléiades orthoimage.**

| Mean zonal Precipitation correction factor (-) | SIA | F2019 | IGM | Snowslide model | Forward modeling |
|---|---|---|---|---|---|
| **Headwalls** | 1.6 +/- 0.9 | 1.7 +/- 1.1 | 1.6 +/- 0.5 | 1.7 | **1.6** |
| **Rognons** | - 0.3 +/- 1.2 | 0.3 +/- 1.4 | - 0.5 +/- 0.8 | 1.0 | **0.7** |
| **Fond du Cirque** | 1.3 +/- 0.3 | 1.2 +/- 0.5 | 1.5 +/- 0.3 | 1.6 | **1.3** |
| **Argentière main** | 0.7 +/- 0.3 | 0.8 +/- 0.5 | 0.7 +/- 0.2 | 1.0 | **1.0** |
| **ALL except Tour Noir** | **1.0 +/- 0.5** | **1.1 +/- 0.7** | **1.0 +/- 0.3** | **1.4** | **1.2** |
| **Tour Noir** | 0.5 +/- 0.2 | 0.5 +/- 0.3 | 0.7 +/- 0.2 | 1.2 | **0.7** |
| **ALL** | **0.9 +/- 0.4** | **1.0 +/- 0.6** | **1.0 +/- 0.3** | **1.3** | **1.1** |

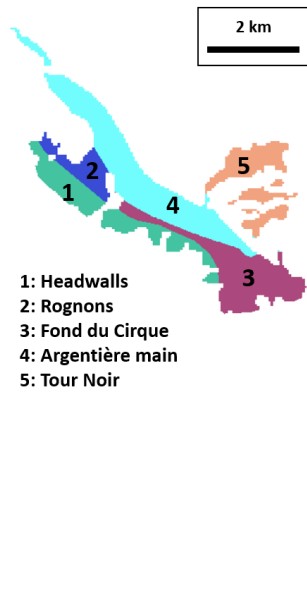

1: Headwalls
2: Rognons
3: Fond du Cirque
4: Argentière main
5: Tour Noir

**Figure 8: Mean zonal precipitation correction factor ($P_{fact}$) calculated after directly differencing the GLACIOCLIM ETI model from the three SMB modelling approaches and the SnowSlide redistribution model for different spatial zones of Argentière**





**Glacier (first four columns, Eq. 8). The last column shows the precipitation correction factor used for the forward modelling, after**
**iteration and convergence of the $P_{fact}$ with the GLACIOCLIM ETI model (section 2.10).**

### 3.4 Glacier projections

Both the GLACIOCLIM ETI and corrected ETI models show a good agreement with the *in situ* SMB measurements
(RMSE<0.71 m w.e. yr$^{-1}$; R$^2$>0.95), despite a lower than expected mass balance gradient on the Tour Noir tributary for the
corrected ETI model (Fig. S13). The final $P_{fact}$ used in the corrected ETI model leads to a higher variability in the
distributed SMB, with relatively high values of accumulation (up to 8 m w.e. yr$^{-1}$) at the base of the headwalls, and lower
values than the GLACIOCLIM ETI for the Tour Noir and lower Rognons tributaries (Fig. S12, S13). Both SMB models
used for historical simulations with the full Stokes model Elmer/Ice show a good agreement with the geodetic and *in situ*
measurements over the period 1907-2022 (Fig. S14a), highlighting almost identical SMB and volume variations (Fig. S14).
From 2020 onwards however, the glacier mass balance of the GLACIOCLIM ETI model becomes more negative than the
corrected ETI model, under the RCP 4.5 climate scenario. This leads to a faster retreat of the Argentière main glacier trunk
(without Tour Noir) and 46% less volume by the end of the century than for the corrected scenario (Fig. 9, S14b). This
increased retreat of Argentière without accounting for avalanching in the GLACIOCLIM scenario is partly compensated by
the Tour Noir tributary retreating slower than with the reduced $P_{fact}$ prescribed in the corrected ETI SMB. When only
accounting for Argentière and the Rognons tributary, the volume is 8%, 25% and 71% lower in 2022, 2053 and 2099,
respectively, for the GLACIOCLIM scenario compared to the corrected scenario.

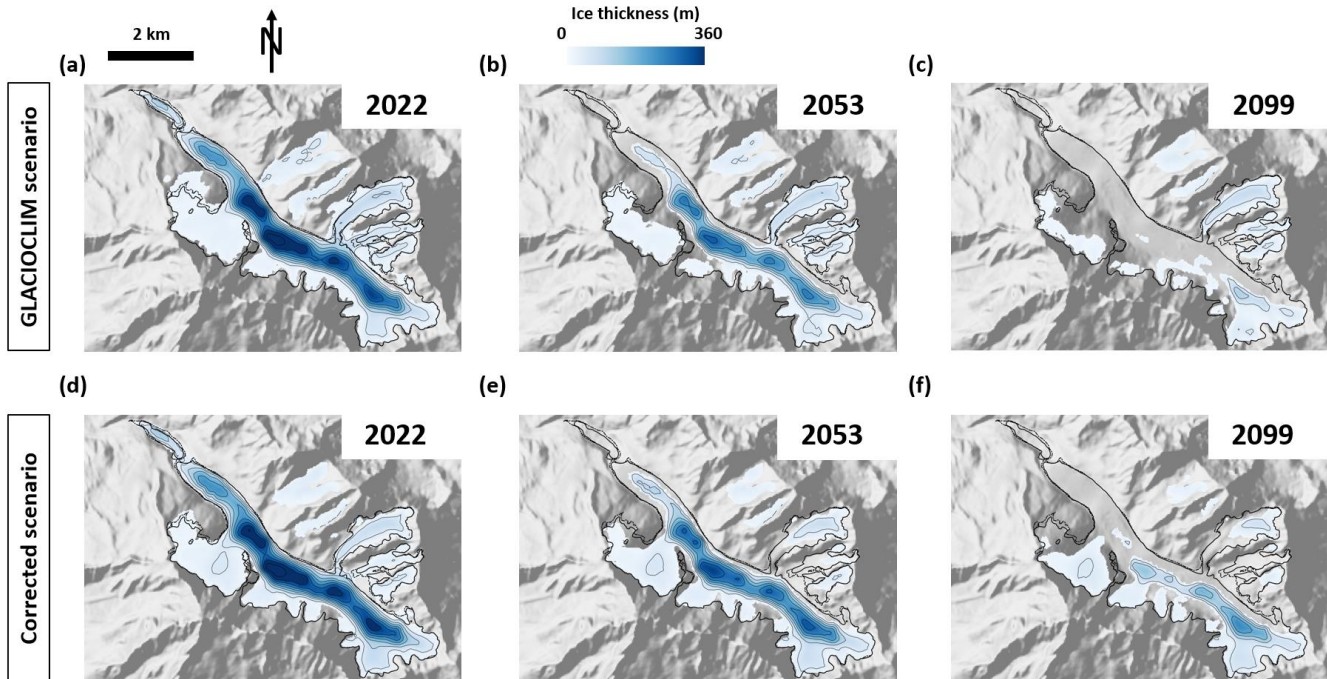





**Figure 9: Distributed ice thickness evolution in the Argentière catchment for the period 2022-2099 using the RCP 4.5 climate scenario for the GLACIOCLIM (a-c) and Corrected (d-f) scenarios. Background is the hillshade of the AW3D30 30 m DEM (Tadono et al., 2014). The black thickness contour lines are spaced every 50 m. The black outlines indicate the glacier outlines manually derived from the 08/09/2020 Pléiades orthoimage, and the dashed outlines show the glacier outlines derived from the 19/08/2012 Pléiades orthoimage.**

## 4 Discussion

### 4.1 Distributed SMB inversion

The distributed SMB simulations from the three modelling approaches show consistent spatial patterns, and a good agreement with the *in situ* measurements (RMSE < 0.96 m w.e. yr$^{-1}$), slightly higher than the values obtained (< 0.6 m w.e. yr$^{-1}$) in the lower ablation zone of Morteratsch and Pers Glaciers from UAV data (Van Tricht et al., 2021) and similar to those obtained on Wolverine Glacier with *in situ* measurements of emergence (RMSE < 0.98 m w.e. yr$^{-1}$; Zeller et al., 2023). These values are still higher than the uncertainty of the *in situ* measurements ranging between 0.15 and 0.30 m w.e. yr$^{-1}$ in the ablation and the accumulation zone, respectively (Thibert et al., 2008; Vincent et al., 2018). We computed the mean value for the full 2012-2021 period, but assuming that the flux divergence and firn density remain constant over ~1 decade, these could be refined to yearly or even seasonal time-scales using high-resolution elevation changes from the Pléiades DEMs (Jourdain et al., 2023; Zeller et al., 2023), with some additional uncertainties caused by the DEM differencing over shorter time periods (Beraud et al., 2023).

For all modelling approaches, the uncertainty in ice thickness is responsible for most of the uncertainties, especially for the tributaries with no ice thickness observations. This highlights the importance of these observations to constrain the mass flux (GlaThiDa Consortium, 2020; Rabatel et al., 2018). The good agreement between the three modelling approaches but larger SMB uncertainties at the margins indicate that, as expected, the absolute thickness has less influence on the flux divergence than the thickness gradient (Cuffey and Patersen, 2010). In the particular case of Argentière Glacier with well-resolved surface velocity from very high spatial resolution Pléiades images, the uncertainties in the velocity field have a limited influence on the final uncertainties, and this also helps constrain the IGM inversion (Fig. S7). To test the sensitivity of our approach to the quality of the velocity data, we conducted some runs to invert the SMB from the F2019 thicknesses obtained with the SGSs and using the surface velocity derived from coarser Sentinel-2 data available at global scale in Millan et al. (2022) following the same Monte Carlo approach, but with an uncertainty on the velocity of 10 m yr$^{-1}$ (Fig. S15). This modelling approach shows a much weaker agreement with the GLACIOCLIM SMB, with an RMSE (R$^2$) of 1.2 m w.e. yr$^{-1}$ (0.63) and 1.3 m w.e. yr$^{-1}$ (0.70) for the Argentière and Tour Noir stakes, respectively (Fig. S12b). This indicates that the velocity fields can in some cases become a major limitation for the calculation of the distributed SMB. The surface velocity from the Pléiades image pairs also indicates a limited glacier slow-down over the study period, particularly in the lower part of the main glacier tongue, which, given the observation bias towards the second half of the study period, could lead to a slight underestimation of the flux divergence in this zone for the selected study period (Fig. S16).



In our processing we assumed that there is no significant change in firn compaction rates, and as such the influence of the density uncertainties on the final uncertainty are of secondary concern (Fig. S9, S10). This assumption may not hold for other glaciers and other time periods, for which changing firn densification may lead to surface lowering with little influence

on the surface mass balance or flux divergence (Belart et al., 2017; Pelto et al., 2019; Réveillet et al., 2021; Vincent et al., 2020; Zeller et al., 2023). Firn compaction could partly explain the low SMB values relative to the Tour Noir GLACIOCLIM measurements in all three modelling approaches. To give an order of magnitude, assuming a 20 m thick firn layer, a 100 kg m$^{-3}$ increase in mean density over the study period would explain a 0.2 m w.e. yr$^{-1}$ SMB difference, which is however insufficient to explain the differences between observed and SIA or F2019 SMB. The SMB values over Tour Noir

are especially low considering that these tributaries are also surrounded by relatively steep headwalls leading to a $P_{fact}$ of 1.2 according to the snow redistribution model (Fig. 8). The best simulations at these locations come from the IGM modelling approach, which indicates that these deviations could also partly be explained by the spatial variability of the sliding coefficient, which is modelled to decrease with elevation for these glaciers (Fig. S7e). Another location with high uncertainties in the calculated SMB is the Rognons tributary, which has high local velocities, and where the calculated SMB

are very low, leading to a final $P_{fact}$ of 0.7 (Fig. 8). This tributary is relatively steep compared to the main glacier trunk, and densely crevassed (Fig. S17). There seems to be a high variability of accumulation at this location, with some crevassed zones remaining snow free in the winter (Fig. S17c-d) and the maximum snowline elevation at the end of the melt period is on average 60 m higher than on the main glacier trunk (Table S2). This could indicate lower accumulation rates at these locations, leading to a longer melt period than at the stake locations, thus a reduced SMB, but also enhanced ablation caused

by crevasses and their increased surface area and turbulent fluxes (Colgan et al., 2016; Zeller et al., 2023). However, the high uncertainties of the calculated SMB and ice thickness at this location could also indicate a discontinuity in the bed elevation, leading to a high local flux divergence that would be difficult to capture with the inversion approaches used.

We also note that an important assumption of the SMB inversion here was to take the density of the ice flux as a constant

equal to 900 kg m$^{-3}$. This is assuming that the firn layer thickness is negligible relative to the ice thickness in every point of the glacier, which is likely not the case, especially near the glacier margins in the accumulation area. A firn core taken at Col du Midi, located ~12 km from Argentière Glacier and at a relatively higher elevation (~3,500 m a.s.l.) indicates a firn density varying between 600 and 800 kg m$^{-3}$ between 5 and 20 m depth (Jourdain et al., 2023). We also expect an important compaction at locations affected by avalanching, but assuming that half of the glacier column is composed of firn with an

average density of 700 kg m$^{-3}$, this would still lead to a systematic 10% reduction in the mass flux and could reduce our inverted precipitation correction factors at the base of the headwalls by a similar amount. This is however not straightforward to correct as this varying density would need to be accounted for first in the ice thickness inversions, which is currently not the case (e.g. Millan et al., 2022).



The smoothing of the flux divergence failed the mass conservation assumption; we resolved this by redistributing the mass excess or shortage homogeneously across the glacier. This glacier-wide mean flux divergence before redistribution was of the order of several tens of centimetres per year for all three modelling approaches (Table S3). The mean value obtained in the Monte Carlo simulations ranged from 0.11 to 0.23 m yr$^{-1}$ and reached 0.73 m yr$^{-1}$ for a particular ice thickness of the SIA inversion (Table S3). Smoothing the thickness and velocity gradient prior to the flux divergence calculation, as suggested

from a study on Morteratsch and Pers Glaciers (Van Tricht et al., 2021), resulted in higher mean values between 0.38 and 0.59 m yr$^{-1}$, which led us to choose to apply the smoothing to the flux divergence only. Interestingly, the regularisation of the flux divergence calculated with centred differences proposed in IGM (Jouvet, 2023; Jouvet and Cordonnier, 2023) still led to mass conservation problems. This was an indicator that a stronger constraint on the glacier-averaged flux divergence could be needed, but when imposing this additional constraint, this came at the expense of increased SMB uncertainties and lower

RMSE values (Fig. S18). In this particular case the homogeneous redistribution of mass seemed to be the better approach.

## 4.2 Attribution to avalanching

The advantage of the distributed SMB relative to the glaciological measurements at point locations on the glacier is that one can more accurately quantify the spatial variability of the SMB, even for a well-studied glacier such as Argentière (Vincent et al., 2018). This spatial variability is much stronger than predicted by the GLACIOCLIM ETI model calibrated with direct

observations, especially in terms of accumulation. In fact, the variability of accumulation predicted by the three modelling approaches and, as a result, the corrected ETI model agree much better with the winter accumulation measured by GPR (Fig. S10), which was also a finding of Zeller et al. (2023). One of our objectives was to use the SMB inversions to be able to constrain the contribution from avalanches to the SMB at the margins of the glacier. The potential hazard at these locations and the highly dynamic mass redistributions make the direct measurement of accumulation on the avalanche cones

complicated (Hynek et al., 2023; Purdie et al., 2015). Using our approach, we can interpret the deviations from our GLACIOCLIM ETI model calibrated at the stake locations as the results of processes unaccounted for in the model. The higher precipitation correction factors at the base of the headwalls coincide with regular avalanche deposits mapped with satellite radar images (Fig. 7a; Kneib et al., 2023). The values obtained are similar to those obtained with the SnowSlide parametrization, for which the precipitation correction factor is 1.7 compared to the values between 1.6 +/- 0.5 and 1.7 +/-

1.1 (Fig. 7b, 8; Bernhardt and Schulz, 2010). This is however not the case for the Tour Noir tributaries, which could also be indicative of other processes contributing negatively to the SMB in compensation of the avalanching. In fact, there are a number of processes unaccounted for in our GLACIOCLIM ETI model which could also explain some of the observed variability. These could be related to the topographic shading at the base of these north-facing headwalls, to the preferential redistribution of snow by wind, to a varying precipitation lapse rate at high elevation along the headwalls, or to the lower

albedo values caused by the snow cover lasting longer on the avalanche cones (Florentine et al., 2018; Olson and Rupper, 2019). Testing most of these hypotheses would likely require more advanced observations or modelling schemes to be able to discriminate between these different contributions (Mott et al., 2019; Voordendag et al., 2024), but our approach at least





provides an estimate of their overall local contribution to the SMB. While the direct mass redistribution from the headwalls is likely not the only process leading to locally high accumulation values, the snow redistribution parametrization seems to
give an appropriate order of magnitude of this contribution. It does not represent individual events, but it gives an overall contribution from all the snow redistribution processes, from snow drifts to large avalanches and serac falls, assuming that the snow and ice content on the headwalls remains constant over time during the study period (Bernhardt and Schulz, 2010; Gruber, 2007).

Our precipitation correction factors are in line with the empirical correction factors set by Gilbert et al. (2023) to reach an agreement between modelled and observed surface velocities. In their study, a precipitation correction factor of 1.4 was applied to a zone similar to our headwall zone. Furthermore, our precipitation correction factors for the Rognons tributary result in a better fit between the modelled and observed surface velocity patterns and glacier extents than what had been achieved in that study (Gilbert et al., 2023; Fig. S19). While previous studies inferred the avalanche contribution at the scale
of entire glaciers (Laha et al., 2017), our distributed SMB product for Argentière offers a very detailed perspective on the spatial variability of the accumulation. It is also directly inferred from remote sensing observations, and as such can be a useful reference for snow redistribution models, which have been applied in several glacio-hydrological studies (Burger et al., 2018; Mimeau et al., 2019; Ragettli et al., 2015). Specific studies on the contribution of avalanches to glacier mass balance have highlighted the importance of this mass redistribution for the accumulation. On Freya Glacier in Greenland, an
important avalanche event in 2018 resulted in the 2013/14 - 2020/21 mass balance reaching +0.25 m w.e. instead of -0.30 m w.e. (Hynek et al., 2023). A modelling exercise for three glaciers of the central Andes also showed that considering avalanches had a similar effect on the glacier evolution than adding 10 cm of debris on the glacier surface (Burger et al., 2018). These results are in line with our own 1907-2100 simulations using Elmer/Ice with the GLACIOCLIM and corrected ETI SMB models. These simulations show that without accounting for avalanches, Argentière Glacier without the Tour Noir
tributary would have 71% less volume by 2100 with the RCP 4.5 climate scenario. In the distributed scenario, the increased accumulation at the base of the Argentière headwalls is partly compensated by the more negative SMB on the Améthyste and Tour Noir tributaries, which explains the agreement between both scenarios for the historical period. However, once these tributaries get fully disconnected from the main glacier trunk, the distributed scenario leads to more ice being maintained at the base of the headwalls (Fig. 9). This highlights the importance of explicitly accounting for avalanches for
this particular glacier. We note that these simulations were conducted with a fixed $P_{fact}$ that does not account for an upward migration of the rain/snow transition, and therefore likely underestimates the future relative contribution of avalanches to the glacier accumulation.

Our estimates of the current and future avalanche contribution to the mass balance of Argentière Glacier are promising, but
show that it remains difficult to directly account for this snow redistribution without considerable uncertainties, given the spatial and temporal variability of these processes (Hynek et al., 2023; Kneib et al., 2023). As such, this study calls for more





detailed observations of mass redistribution from avalanches, on or off-glacier (Hynek et al., 2023; Sommer et al., 2015), and for a better representation of these processes in glacier models. More generally, our study showcases the use of the SMB inversion to identify ablation or accumulation hotspots, which may not be detectable by a network of *in situ* measurements, even on a well-known glacier such as Argentière. Such an approach can therefore provide crucial information on the variability of the SMB of mountain glaciers to target local processes such as avalanching or sub-debris melt (McCarthy et al., 2022; Rounce et al., 2018). It also has the opportunity to be expanded to the regional scale (Cook et al., 2023; Miles et al., 2021), with the main current limitation being the quality of the surface velocity observations, especially in the accumulation zones.

## 5 Conclusion

Our study leveraged high-resolution remote sensing data to invert for the distributed SMB of Argentière Glacier using three different methods to estimate the ice thickness and calculate the ice flux and its divergence. These approaches displayed consistent patterns of SMB and showed a good agreement with measurements at stakes, with RMSE values lower than 0.96 m w.e. yr$^{-1}$. They highlighted a strong spatial variability in SMB, much higher than what would be expected from the GLACIOCLIM ETI model that was calibrated against the stake measurements.

This variability can be at least partly attributed to avalanching from the headwalls on the left-hand side of Argentière Glacier. This location is characterised by regular large avalanches that are visible in satellite radar images, and a simple parametrization of snow redistribution on the headwalls also indicates high accumulation rates at the base of these headwalls. Based on the SMB inversions we estimated that this contributed to an additional 60% mass accumulation at these locations, which was equivalent to an additional 20% mass accumulation at the scale of Argentière Glacier, without the Tour Noir and Améthyste tributaries.

We used these distributed SMB inversions to propose a corrected SMB model which accounts for this additional mass accumulation. This leads to twice more mass being conserved by 2100 in an RCP 4.5 climate scenario, and to a slower retreat of Argentière Glacier at the base of the headwalls. Our results therefore highlight the role of avalanches for the mass balance and future evolution of Argentière Glacier, and the importance of accounting for this effect in glacier models. More generally, it showcases the potential of SMB inversions to derive key information on the spatial variability of the surface mass balance, and to attribute this variability to specific processes.

## Code availability

Codes to derive the surface mass balance are available on GitHub: https://github.com/miles916/grid_continuity_SMB. Codes for the SIA thickness inversion and the gaussian simulations of the ice thickness are also available online: https://github.com/adehecq/argentiere_pleiades_smb. We used the IGM model for one of the ice thickness modelling approaches: https://github.com/jouvetg/igm and the Elmer/Ice model for the forward modelling: https://github.com/elmercsc/elmerfem. We have also made the SnowSlide snow redistribution scheme available online: https://github.com/OGGM/Snowslide.

## Data availability

Elevation change, velocity, ice thickness and distributed SMB products will be made available on Zenodo upon acceptance of the manuscript. All *in situ* mass-balance data are available in the GLACIOCLIM database at https://glacioclim.osug.fr/ and through the World Glacier Monitoring Service (WGMS) database.

## Competing interests

The authors declare that they have no conflict of interest.

## Acknowledgements

This project has received funding from the Swiss National Science Foundation (SNSF) under the Postdoc.Mobility programme, grant agreement P500PN_210739, CAIRN, "Contribution of avalanches to glacier mass balance". The Pléiades images used in this study were obtained through the Kalideos-Alpes project (https://alpes.kalideos.fr) funded by the French Space Agency (*Centre National d'Etudes Spatiales*, CNES) and the DINAMIS initiative through the research infrastructure DATA TERRA (https://dinamis.data-terra.org/).

Authors from IGE acknowledge the support from the LabEx OSUG@2020 (*Investissements d'Avenir* - ANR10 LABX56). The authors thank the GLACIOCLIM monitoring service (CNRS-INSU, UGA-OSUG, IRD, INRAE, IPEV, Météo France) for the *in situ* surface mass balance measurements (https://glacioclim.osug.fr/).

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
