# Peer review of "Distributed surface mass balance of an avalanche-fed glacier"

_EGUsphere, 2024_

## Author Comment (AC1)

**Reviewer 2**

The manuscript by "Kneib et al." describes and demonstrates the calculation of a Surface Mass Balance (SMB) field for the Argentière Glacier in the Mont Blanc massif using high-resolution dh/dt fields (from Pleiades DEMs) and the calculation of ice flux divergence (using three different ice thickness datasets, a high-resolution velocity field, and two different methods to calculate the divergence). The resulting SMB field is then compared with results from an SMB model and stake observations, with the ultimate goal of obtaining a precipitation correction that better represents avalanche deposits. The impact of this correction on the glacier's future ice volume is subsequently examined using a 3D thermomechanical ice flow model.

The manuscript is very well-written, with a clear sequence of the steps undertaken by the authors. The reader can easily follow the process, which allows to apply a similar setup for other glaciers (highly appreciated). The various figures, both in the text and in the supplementary material, are well-chosen to represent the results obtained. In addition to the very detailed work carried out (including all uncertainties), the result is compelling and demonstrates the significant added value of the method.

In my opinion, this paper is definitely a valuable contribution to the glaciological community and is therefore highly recommended for publication in 'The Cryosphere.'

We would like to thank Dr Van Tricht for his thorough review and his very relevant and constructive comments.

I have only a few textual comments and a couple of minor clarifications which I would like to see in the revised version. The only more substantial work I can see is calculating/presenting a mass balance derivation for each year in the studied period (2012-2021). How well does the mean specific mass balance obtained with the applied method represent the mean specific mass balance from the glaciological method and SMB model? And how does the mass balance of the stakes per year compare with the method used?

Thanks for this suggestion. There were several reasons why we choose this relatively long time period to conduct our SMB inversions, which are mainly due to uncertainty considerations:

- When reducing the observation period to time intervals shorter than 3 years it has been shown that at the glacier scale the density conversion factors could deviate outside the range of 850 +/- 60 kg/m$^3$ at the glacier scale (Huss, 2013). To stay within reasonable uncertainty bounds, particularly in the accumulation zones, we therefore chose to stick to a longer time series.
- Elevation change and velocity data become very gappy at the annual scale, despite the quality of the Pléiades images. These gaps, especially at the base of the headwalls (which are subject to steep slopes and topographic shadows that affect the velocity and elevation data) need to be filled for our inversion approach and this may lead to artefacts and inconsistencies in the products. Figure S16

show examples of velocity maps for the periods 2012-2015, 2015-2018 and 2018-2021 which indicate a high density of gaps at the base of the headwalls, particularly for the period before 2018. Aggregating these velocity maps is therefore crucial to reduce the uncertainty in the velocity product. Similarly, one can refer to the Figure 2 in Beraud et al. (2023) (see copy below) that indicates the numbers of gaps needing to be interpolated in the Argentière DEMs. While their study also includes winter pairs, this shows that some locations on the glacier are particularly difficult to map, thus the need to use as many DEMs as possible to constrain the elevation change patterns in these locations.

[Figure]

**Fig. 2 from Beraud et al. (2023).** Left: Map of the Glacier d'Argentière showing the number of interpolations between consecutive DEMs (total 11 pairs). Right: Snow-free stable areas used by three co-registrations or more. Spring areas nearly fully cover summer areas.

- We also expect a high temporal variability in accumulation (especially from avalanching, Hynek et al., 2024), as can be seen in the elevation change patterns shown in Figure 4 of Beraud et al. (2023). While it is definitely very interesting to analyse this annual variability, we were more interested in the average contribution of avalanches to the surface mass balance over a time period long enough to smooth out this variability and analyse the long term signal.

For all these reasons we decided to stick to our relatively long study period of 2012-2021, which uses all the data that was available from the Pléiades observations. We agree that it could have been possible to reduce this period by a few years or cut it in two without increasing the uncertainties too much, but this was not necessary for the purpose of our research questions and therefore felt beyond the scope of this study. We will therefore keep it as it is. These points are already mentioned in the discussion section:

'We computed the mean value for the full 2012-2021 period, but assuming that the flux divergence and firn density remain constant over ~1 decade, these could be refined to yearly or even seasonal time-scales using high-resolution elevation changes from the Pléiades DEMs (Jourdain et al., 2023; Zeller et al., 2023), with some additional uncertainties caused by the DEM differencing over shorter time periods (Beraud et al., 2023).'

Specific comments:

- Line 15: "Particularly" might be removed here

  Will be removed as suggested

- Line 16: High resolution (<-> high quality) as well?

  We will add it as suggested

- Line 18: A bit unclear if the approach to invert only uses three different ice thickness estimates or three different methods. Further, if formulated like now, it seems that the ice thickness setups show a good agreement, but it is the three different inversions that do show the good agreement.

  Agreed. We will change this as follows:

  'Three inversions are conducted using three different ice thickness modelling approaches, two of which are constrained by observations. The inversions all show a very good agreement between inverted surface mass balance and *in situ* measurements (RMSE between 0.50 and 0.96 m w.e. yr$^{-1}$ for the 11-year average).'

- Line 19: After reading, the "consensus F2019 estimate" is not constrained by the ice thickness measurements?

  See response to comment above.

- Line 20: Maybe mention the range of RMSE values?

  Will be added as suggested.

- Line 21: "the" modelling approaches

  Will be modified as suggested.

- Line 23: "Avalanching" -> avalanche deposits

  Will be modified as suggested.

- Line 33: Accumulation zone

  Will be modified as suggested.

- Line 36: and extrapolation?

  Will be added as suggested

- Line 41: Compaction as well?

  We considered compaction to be an 'internal process' but will add it as an example: 'This mismatch is due to internal processes such as compaction'

- Line 45: Remove "and"

  Will be removed.

- Line 52-53: Potentially add a reference to "Turchaninova A.S., Lazarev A.V., Marchenko E.S., Seliverstov Y.G., Sokratov S.A., Petrakov D.A., Barandun M., Kenzhebaev R., Saks T. Methods of snow avalanche nourishment assessment (on the example of three Tian Shan glaciers). Ice and Snow. 2019;59(4):460-474. https://doi.org/10.15356/2076-6734-2019-4-438

Really nice reference, thanks! We will add it here.

- Line 67: "ice flux"

  Will be modified as suggested.

- Line 67: Therefore -> subsequently

  Will be modified as suggested.

- Line 81: "which" does not have a direct link here. You probably mean that ice thickness and velocity is less constrained in the accumulation zone and therefore the uncertainty of the product is larger, butt this might need to be a bit clearer stated

  We will modify the sentence for clarity: 'However, these estimates depend on the quality of the ice thickness, velocity and elevation change data which are less constrained and therefore lead to higher uncertainties in the accumulation area of the glaciers (Miles et al., 2021).'

- Figure 1: The colour scale is a bit large for the values of the SMB (especially the positive ones). Maybe limit this to +3 m w.e. yr-1 so that you can see a bit more the variations? The circles of the GPR locations are too large so that you cannot locate the individual points (it now seems to be more a line of GPR). Panel b, is it an option to zoom in somehow (e.g., to the Mont Blanc massif)?

  Thanks for these suggestions. We will use an asymmetrical color scale to constrain it to [0; 3] m w.e. yr-1 for the positive values. The GPR acquisitions are very dense so they really are more lines than points, we will change the symbol in the legend. We will also zoom panel b to the Mt. Blanc massif.

- Line 132: Which software is used to compute the surface displacement, velocity?

  The processing chain used is the same as the one described in Millan et al. (2019) and more recently in Mouginot et al. (2023), which uses Python 3.7 and Fortran. We will specify this in the text:

  'The velocity fields were obtained using normalised cross-correlation and the images were co-registered using the median velocity of the off-glacier terrain using the workflow described by Millan et al., (2019) and Mouginot et al. (2023).'

  Mouginot, J., Rabatel, A., Ducasse, E., & Millan, R. (2023). Optimization of Cross Correlation Algorithm for Annual Mapping of Alpine Glacier Flow Velocities; Application to Sentinel-2. *IEEE Transactions on Geoscience and Remote Sensing, 61*, 1–12. https://doi.org/10.1109/TGRS.2022.3223259

- Line 141: You mention later that the F2019 did not use the GPR measurement, so it is a bit strange to mention here that the three methods use in situ data

  We will modify the sentence as follows to clarify this point: 'We used distributed ice thicknesses obtained from three different modelling approaches, two of which are constrained by *in situ* ice thickness observations.'

- Line 184: There is some error between ice thickness (H) change (dH/dt) and surface elevation (h) change (dh/dt).

  We will correct this for equations (2) and (3).

- Line 184: Formulating $H_2O$ makes the equation for me a bit confusing. Consider writing just $r_w$ and $r_i$.

  We will change rH2O to rw

- Line 185: Any evidence that internal and basal mass balance are negligible from previous studies?

  The good agreement between geodetic mass balance and surface mass balance measurements shown in Beraud et al. (2023) seems to comfort this hypothesis. Basal ablation from geothermal heat is generally very limited on temperate alpine glaciers (Alexander et al., 2011). Neglecting the internal mass balance is a stronger hypothesis, which is discussed in more details in the discussion section:

  'In our processing we assumed that there is no significant change in firn compaction rates, and as such the influence of the density uncertainties on the final uncertainty are of secondary concern (Fig. S9, S10). This assumption may not hold for other glaciers and other time periods, for which changing firn densification may lead to surface lowering with little influence on the surface mass balance or flux divergence (Belart et al., 2017; Pelto et al., 2019; Réveillet et al., 2021; Vincent et al., 2020; Zeller et al., 2023). Firn compaction could partly explain the low SMB values relative to the Tour Noir GLACIOCLIM measurements in all three modelling approaches. To give an order of magnitude, assuming a 20 m thick firn layer, a 100 kg m$^{-3}$ increase in mean density over the study period would explain a 0.2 m w.e. yr$^{-1}$ SMB difference, which is however insufficient to explain the differences between observed and SIA or F2019 SMB.'

  We will add the two references mentioned above to the sentence to justify this choice.

  Alexander D, Shulmeister J, Davies T. High basal melting rates within high-precipitation temperate glaciers. *Journal of Glaciology*. 2011;57(205):789-795. doi:10.3189/002214311798043726

- Line 188: Here you refer to ice thickness as H

  Apologies for this, we will change it in equation 3.

- Line 211: Any estimate for y based on observations or modelling?

  To our knowledge, no study has focused on distributed estimates of '*y*' for Argentière Glacier, which is why we derived it from the IGM simulations (Fig. S7). There is a preprint in EGUsphere that has looked at ice deformation using borehole data in the lower part of Argentière Glacier (Roldan-Blasco et al., 2024), finding a yearly mean vertical velocity of 38 m/yr for a surface velocity of 43 m/yr, so a 0.88 y ratio. However, these are local measurements in the lower part of the ablation area and likely not representative of the entire glacier. We therefore considered a uniform distribution of y between 0.8 and 1 to quantify its uncertainties in a conservative way (Cuffey & Paterson, 2010).

- Line 215: If formulated like this, it seems you calculate the ice flux divergence in three different ways. But as far as I understand, this is not the case? Three different ice thickness estimates are used, and two different approaches to determine the ice flux divergence.

Yes, good point, we will make this more explicit: 'We computed the flux divergence using three different ice thickness estimates and two different approaches, leading to three SMB estimates (Eq. 2), the **SIA, F2019** and **IGM** estimates.'

- Line 223 and 224: Some repetition here of the ranges for the uncertainty (with lines 208-213).

  We will remove this sentence to avoid repetition.

- Figure 2: Like for F1, is it possible to limit the range of the colour scale for the SMB?

  We will use an asymmetrical color scale to constrain it to [0; 3] m yr-1 for the positive values.

- Line 243: I do not completely get this 10% best SMB estimates. Can you clarify this a bit?

  Yes, apologies if that was not clear. The idea here was to take the best 10% scenarios (relative to in situ measurements) to be used as a reference to quantify the spatial variability and contribution from avalanches. We will clarify it at the end of the sentence:

  'We compared the inverted distributed SMB with the *in situ* SMB at the GLACIOCLIM stake locations and for each modelling approach selected the best 10% SMB estimates that minimised the weighted quadratic sum of the RMSE of the Argentière and Tour Noir stakes to be used as reference scenarios for the quantification of the avalanche contribution.'

  This complements the details given in 2.9:

  'Here, we re-evaluated this over the Rognons, Tour Noir and Améthystes tributaries and in the accumulation area using the mean of the best 10% inverted SMB patterns of each modelling approach'

- Line 273: How does the exponentially decay of the albedo is determined? Which time scale is used?

  This exponential decay is based on the parametrization presented in Hock and Holmgren (2005), using the same parameters as in their study, with a decay of -0.1 day-1. We will add this reference here:

  'Where $\alpha$ is the local surface albedo (=alpha_ice or =alpha_snow, which decreases exponentially with the age of the surface snow; Hock and Holmgren, 2005), '

- Line 341: Again, here you mean dh/dt I guess? The difference between dH/dt and dh/dt is not clear throughout the manuscript. I guess because bedrock elevation is considered to be stable, both are the same, but you should state this somewhere and from then on work always with dH/dt

  We will change this to dh/dt.

- Line 343: The median of the 2012-2021 period for every grid cell?

  Correct, we will specify this here.

- Figure 3: Both for the elevation change and the surface velocity, the colour scales could be optimized (wider for velocity, smaller for elevation change).

We will modify this as suggested.

- Line 348: Median or mean velocity (<-> line 343)?

Good catch, this is the median, we will correct it in the caption.

- Figure 5: You state different modelling approaches, but in fact it concerns two different ice flux modelling approaches and three different ice thickness estimates?

For clarity we will change this sentence to: '(a) SIA, (b) F2019 and (c) IGM SMB estimates.'

- How well can you invert the annual surface mass balance? There is only a focus on the multi-year average

Please refer to our answer to the general comment above.

- Line 385-389: I do not completely get this sentence (which I find too long).

We will cut it into two different sentences for clarity and simplify the second part: 'There is a good agreement with the SMB measurements on Argentière for both the SIA and F2019 thickness with RMSE values lower than 0.67 m w.e. $yr^{-1}$ (Fig. 6d-e). Uncertainties of the F2019 and SIA estimates reach up to +/- 1.2 m w.e. $yr^{-1}$ (F2019) and +/- 0.8 m w.e. $yr^{-1}$ (SIA) at the stake locations and higher at the margins of the glacier, particularly over the Rognons tributary (Fig. S6), where they locally reach up to +/- 6-8 m w.e. $yr^{-1}$'

- Figure 9: Very cool to see the impact of taking into account the avalanches. I wonder, however, if it is possible to show the flowlines. My first guess would be that the ice at the location of the avalanche deposits does not flow to the central trunk but moves along the headwall. But this must be different because the central trunk maintains more ice?

Indeed, according to our simulations with Elmer/Ice, initially the ice at the location of the avalanche deposits does not flow to the central trunk but moves along the headwall (Fig. R1). However, as the glacier retreats the geometry changes and the flow from the avalanche cones does converge into the main glacier trunk.

[Figure]

*Figure R1: Glacier flowlines in 2021 for the corrected scenario.*

- Line 558: Is shading not included in the simplified energy balance model? By modifying the incoming radiation?

  We did not include shading in our simplified energy balance model to reduce the computation cost. This decision is based on preliminary tests that we conducted and that showed that including shading had a limited effect on the surface mass balance as: 1/ in winter the melt is very limited even in the non-shaded locations and 2/ in summer the sun is high enough in the sky that the large majority of the glacier surface receives the same amount of SW radiation. We do agree however that in some cases topographic shading could lead to strong mass balance gradients, and that it will be worthwhile to account for this effect in glacier models.

- Line 603: I guess also the ice thickness is crucial when the approach would be applied on larger scales? As you show in the sensitivity analysis

  It is true that in our sensitivity analysis (Fig. S9-S10) ice thickness is responsible for most of the uncertainty. However, this is a particular case as we have very high-quality velocity data for our study. In the end all three ice thickness scenarios seem to perform relatively well, but when we changed to a coarser resolution velocity product (Fig. S15) this really had a strong negative impact on the results of the SMB inversion. This is why we have stated that velocity seems to be the main current limitation. But of course, this would need to be tested at other sites. We will also mention ice thickness here: 'with the main current limitations being the quality of the surface velocity observations, especially in the accumulation zones, followed by the availability of ice thickness measurements to constrain distributed ice thickness estimates.'

- Line 606: cf line 16 high-resolution vs high-quality (both are true…)

  Good point, we will add 'high quality' here.

---

## Author Comment (AC2)

We'd like to thank both reviewers for their very valuable inputs and thorough reviews. Our answers to each comment are indicated in blue below.

**Reviewer 1**

In this paper the authors compare the distributed SMB of well-studied Argentière Glacier calculated (1) by inverting an ice flow model and (2) using an enhanced temperature-index (ETI) mass balance model. With method (1) they find a higher spatial variability of SMB, especially higher accumulation rates at the orographic left side of the glacier, which they attribute mainly to regular avalanching from adjacent steep headwalls. This is supported by the indication of avalanches in radar images and a conceptual snow redistribution scheme. To better describe the spatial variability of the SMB and especially account for the total effect of snow redistribution – in which avalanches are shown to be an important factor - the authors come up with zonal precipitation correction factors. Using these correction factors in the ETI-SMB model they find that the projected glacier volume by 2100 based on RCP 4.5 is higher than without including this effect and that more mass is conserved in the zones below the steep headwalls. Finally they conclude that SMB inversions have a high potential in deriving the spatial variability of the SMB.

We would like to thank Reviewer 1 for their high-quality review, and their very relevant and constructive comments.

General comment:

This is a very interesting study, which is building up on a lot of previous work, taking advantage of the solid data base of Argentière Glacier and advanced glacier modelling. The authors take an impressive modelling effort to calculate the distributed mass balance of Argentiére Glacier and to better constrain the effect of avalanches on the mass balance. Especially the uncertainty analysis using three different ice thickness distributions and their effect on the inverted surface mass balance is very interesting. This study is an important contribution to better quantify the spatial variability of surface mass balance on a glacier and to attribute the observed variability to individual processes.

The paper is well written, well structured and generelly pleasant to read. Methods and results are described in a comprehensible manner, holding a good balance between detailness and readability. The supplement is very usefull in following the details of the analysis and the results. The authors discuss uncertainties and limitations of their study extensively and also put their findings in relation to the relevant literature.

I have only some minor comments on the manuscript.

Minor comments:

L27: you could add the 20% of total mass input for the whole glacier as you did in the conclusions, as this is also a main quantitative finding.

True, we will add this here:

'indicating an additional 60% mass input relative to the accumulation from solid precipitation at these specific locations, which was equivalent to an additional 20% mass accumulation at the scale of Argentière Glacier without its two smaller tributaries'

L118: What do you gain by using 13 DEMs (of every year) and not just the DEMs of the beginning and end of the study period? Signal to noise ration should be best by using those two only. Could you add a line to explain why you are doing that.

The goal here was mainly to reduce the number of gaps in the accumulation area, especially at steep locations or areas affected by shadows (which is the case for a large area at the base of the headwalls, where the avalanche deposits are mostly located). This approach also has the advantage of smoothing out the signal of individual avalanches (or crevasses, or any local process), which could introduce a local bias in one of the DEMs and therefore create an artefact in the dh maps. We will specify this in the corresponding paragraph:

'This approach helps reduce the proportion of gaps in steep locations or areas affected by shadows, and smooths out the signal from individual avalanche deposits.'

L141: As I understand it, only two of the three different ice thickness modelling approaches are constrained by measurements at Argentière Glacier, while the Farinotti (2019) model is contrained by data from a also lot of other glaciers? Maybe you can be more specific here. Besides it would also be interesting to add a line, what was the intention to choose these 3 approaches? I guess to cover the uncertainty, that is introduced by the uncertainty in ice thickness distribution, but maybe there was also the idea to have different model complexities or applicability for glaciers without GPR measuements?

Yes, the F2019 approach does not use the Argentière GPR data but rather has been calibrated with lots of other measurements from all around the world. In this sense the first sentence of this paragraph can be confusing - the intention here was also to say that the SGSs applied to the F2019 were using the Argentière in situ data. We will change this, by mentioning that only 2 of the modelling approaches are constrained by the *in situ* GPR data:

'We used distributed ice thicknesses obtained from three different modelling approaches, two of which are constrained by *in situ* ice thickness observations.'

We will also specify that for the F2019 consensus model, these were calibrated with data from many different sites:

'**The F2019 thickness** estimate from the global product by Farinotti et al. (2019) which was originally derived from five different estimates of various sources, constrained by a large amount of GPR data from all around the world. This is a reference product that is available for all mountain glaciers in the world, and which did not use the GPR measurements made on Argentière.'

The primary goal of using these three different products was indeed to cover the uncertainty coming from different scenarios of thickness distribution. And indeed, this

also enabled us to test thicknesses from different model complexities for a possible transferability to the larger scale. We will add this sentence:

'These three approaches were chosen to encompass the uncertainty in ice thickness as well as to test the influence of model complexity.'

L373 – L376 Here you refer to different locations on the glacier by giving altitudes. Please label some contour lines in Fig.5, so that these locations can be identified faster by the reader.

Good point, we will add these in Figure 5.

L456 Here you write "46% less volume by the end of the century than for the corrected scenario". In L 459 (and also in L585) you write 71% lower in volume, obviously both values without the Tour Noir. This seems inconsistant to me. In the conclusion L620 you write: "twice as much mass being conserved by 2100", which seems to be constitant with the 46% of L456. Maybe try to use the same measures throughout the paper.

Thanks for pointing that out. The 46% actually include the Tour Noir tributaries, while 71% stands for the difference without these. To avoid any confusion, we will rephrase this sentence:

'This leads to a faster retreat of the Argentière main glacier trunk and 46% less volume by the end of the century (with Tour Noir) than for the corrected scenario (Fig. 9, S14b).'

L546 I guess text refers to Fig. S1 and not Fig. S10

Good catch, this should actually be Fig. S8. We will change it accordingly.

L612 This location… perhaps say "this area"

Will be changed as suggested.

L615 "this" perhaps say "this process" or "avalanches"

We will change 'this' to 'avalanches'

Fig.3: For a faster readability perhaps add to the legend: 2020 and 2012 glacier outlines and stable terrain. Consider a color scheme that shows more details, especially in the elevation changes (a).

We will move these elements from the caption to the legend. We will use an asymmetrical colormap to make the positive changes in the elevation change more visible.

Fig S16(d): the distance along centerline is oriented from the snout upwards I guess. Please specify that in the x-axis label or in the Figure Caption.

Indeed, we will modify the x-axis label accordingly.

Fig. 3,5,7,9 and also some Figs in the supplement: Maybe you can avoid repeating the phrase "The black outines indicate the glacier ourlines manually dreived from the … Pléiades.." by just stating the year of the glacier outline in the legend of the figures.

Agreed. Will be modified as suggested.

Maybe I missed it: Which ice thickness distribution did you use for the foreward modelling?

For the forward modelling we used the same thickness distribution as Gilbert et al. (2023), which is an inversion applied with Elmer-Ice and constrained by the GPR measurements. We will specify this: 'The thickness inversion for the forward modelling uses Elmer/Ice, as in Gilbert et al. (2023). '

---

## Author Response (AR1)

Dear Marin Kneib,

Thank you for your comments to the reviewers. The two reviewers suggest your paper is well-written and investigates the distributed mass balance of Argentiére Glacier in detail. They also provide some comments to further improve the quality of your paper. I agree with these comments and encourage you to take into account these comments in the revised paper.

Best,

Dr. Kang Yang
Editor, The Cryosphere

Dear Prof. Yang,

Many thanks for the consideration of our manuscript. We have received two well-informed reviews which are overall very positive on the quality of the analysis and relevance of the science presented in this manuscript. The two reviewers had a number of minor comments related to some of the text and figures which we have now thoroughly addressed. No major change to the content or structure of the manuscript was made, nor was any figure removed or added in the main text or the SI. Our codes and data are now available publicly online and we have updated the Code & Data availability sections accordingly.

Our answers to each specific comment are further indicated in blue below and the line numbers indicated correspond to the Manuscript with tracked changes. We think that the manuscript has been strengthened by these revisions, but none of our main results or conclusions have changed.

Thank you for your consideration of our revised manuscript, which we hope is now acceptable for publication.

Kind regards,

Marin Kneib and Co-authors

We'd like to thank both reviewers for their very valuable inputs and thorough reviews. Our answers to each comment are indicated in blue below.

**Reviewer 1**

In this paper the authors compare the distributed SMB of well-studied Argentière Glacier calculated (1) by inverting an ice flow model and (2) using an enhanced temperature-index (ETI) mass balance model. With method (1) they find a higher spatial variability of SMB, especially higher accumulation rates at the orographic left side of the glacier, which they attribute mainly to regular avalanching from adjacent steep headwalls. This is supported by the indication of avalanches in radar images and a conceptual snow redistribution scheme. To better describe the spatial variability of the SMB and especially account for the total effect of snow redistribution – in which avalanches are shown to be an important factor - the authors come up with zonal precipitation correction factors. Using these correction factors in the ETI-SMB model they find that the projected glacier volume by 2100 based on RCP 4.5 is higher than without including this effect and that more mass is conserved in the zones below the steep headwalls. Finally they conclude that SMB inversions have a high potential in deriving the spatial variability of the SMB.

We would like to thank Reviewer 1 for their high quality review, and their very relevant and constructive comments.

General comment:

This is a very interesting study, which is building up on a lot of previous work, taking advantage of the solid data base of Argentière Glacier and advanced glacier modelling. The authors take an impressive modelling effort to calculate the distributed mass balance of Argentiére Glacier and to better constrain the effect of avalanches on the mass balance. Especially the uncertainty analysis using three different ice thickness distributions and their effect on the inverted surface mass balance is very interesting. This study is an important contribution to better quantify the spatial variability of surface mass balance on a glacier and to attribute the observed variability to individual processes.

The paper is well written, well structured and generally pleasant to read. Methods and results are described in a comprehensible manner, holding a good balance between detailness and readability. The supplement is very usefull in following the details of the analysis and the results. The authors discuss uncertainties and limitations of their study extensively and also put their findings in relation to the relevant literature.

I have only some minor comments on the manuscript.

Minor comments:

L27: you could add the 20% of total mass input for the whole glacier as you did in the conclusions, as this is also a main quantitative finding.

True, we have added this here (L26-28):

'indicating an additional 60% mass input relative to the accumulation from solid precipitation at these specific locations, which was equivalent to an additional 20% mass accumulation at the scale of Argentière Glacier without its two smaller tributaries'

L118: What do you gain by using 13 DEMs (of every year) and not just the DEMs of the beginning and end of the study period? Signal to noise ration should be best by using those two only. Could you add a line to explain why you are doing that.

The goal here was mainly to reduce the number of holes in the accumulation area, especially at steep locations or areas affected by shadows (which is the case for a large area at the base of the headwalls, where the avalanche deposits are mostly located). This approach also has the advantage of smoothing out the signal of individual avalanches (or crevasses, or any local process), which could introduce a local bias in one of the DEMs and therefore create an artefact in the dh maps. We have specified this in the corresponding paragraph (L126-127):

'This approach helps reduce the proportion of gaps in steep locations or areas affected by shadows, and smooths out the signal from individual avalanche deposits.'

L141: As I understand it, only two of the three different ice thickness modelling approaches are constrained by measurements at Argentière Glacier, while the Farinotti (2019) model is contrained by data from a also lot of other glaciers? Maybe you can be more specific here. Besides it would also be interesting to add a line, what was the intention to choose these 3 approaches? I guess to cover the uncertainty, that is introduced by the uncertainty in ice thickness distribution, but maybe there was also the idea to have different model complexities or applicability for glaciers without GPR measuements?

Yes, the F2019 approach does not use the Argentière GPR data but rather has been calibrated with lots of other measurements from all around the world. In this sense the first sentence of this paragraph can be confusing - the intention here was also to say that the SGSs applied to the F2019 were using the Argentière in situ data. We have changed this, by mentioning that only 2 of the modelling approaches are constrained by the *in situ* GPR data (L144-145):

'We used distributed ice thicknesses obtained from three different modelling approaches, two of which are constrained by *in situ* ice thickness observations.'

We have also specified that for the F2019 consensus model, these were calibrated with data from many different sites (L157-160):

'**The F2019 thickness** estimate from the global product by Farinotti et al. (2019) which was originally derived from five different estimates of various sources and constrained by a large amount of GPR data from all around the world. This is a reference product that is available for all mountain glaciers in the world, and which did not use the GPR measurements made on Argentière.'

The primary goal of using these three different products was indeed to cover the uncertainty coming from different scenarios of thickness distribution. And indeed, this

also enabled us to test thicknesses from different model complexities for a possible transferability to the larger scale. We have added this sentence (L145-146):

'These three approaches were chosen to encompass the uncertainty in ice thickness as well as to test the influence of model complexity.'

L373 – L376 Here you refer to different locations on the glacier by giving altitudes. Please label some contour lines in Fig.5, so that these locations can be identified faster by the reader.

Good point, we have added these altitude labels in Figure 5.

L456 Here you write "46% less volume by the end of the century than for the corrected scenario". In L 459 (and also in L585) you write 71% lower in volume, obviously both values without the Tour Noir. This seems inconsistant to me. In the conclusion L620 you write: "twice as much mass being conserved by 2100", which seems to be constitant with the 46% of L456. Maybe try to use the same measures throughout the paper.

Thanks for pointing that out. The 46% actually include the Tour Noir tributaries, while 71% stands for the difference without these. To avoid any confusion, we have rephrased this sentence (L460-461):

'This leads to a faster retreat of the Argentière main glacier trunk and 46% less volume by the end of the century (with Tour Noir) than for the corrected scenario (Fig. 9, S14b).'

L546 I guess text refers to Fig. S1 and not Fig. S10

Good catch, this was actually Fig. S8. We have changed it accordingly (L552).

L612 This location… perhaps say "this area"

Changed as suggested (L616).

L615 "this" perhaps say "this process" or "avalanches"

Changed 'this' to 'avalanches' (L618)

Fig.3: For a faster readability perhaps add to the legend: 2020 and 2012 glacier outlines and stable terrain. Consider a color scheme that shows more details, especially in the elevation changes (a).

We have moved these elements from the caption to the legend. We have also used an asymmetrical colormap to make the positive changes in the elevation change more visible in (a). We also increased the color scale to [0 100] m/yr for the velocity data in (b).

Fig S16(d): the distance along centerline is oriented from the snout upwards I guess. Please specify that in the x-axis label or in the Figure Caption.

Indeed, we have modified the x-axis label accordingly: 'distance from terminus along centreline (m)'

Fig. 3,5,7,9 and also some Figs in the supplement: Maybe you can avoid repeating the phrase "The black outines indicate the glacier ourlines manually dreived from the … Pléiades.." by just stating the year of the glacier outline in the legend of the figures.

Agreed. We have moved this to the caption for figures 3, 5, 7, 9, S6 and S12.

Maybe I missed it: Which ice thickness distribution did you use for the foreward modelling?

For the forward modelling we used the same thickness distribution as Gilbert et al. (2023), which is an inversion applied with Elmer-Ice and constrained by the GPR measurements. We have specified this L326-327: 'The thickness inversion for the forward modelling uses Elmer/Ice, as in Gilbert et al. (2023).'

**Reviewer 2**

The manuscript by "Kneib et al." describes and demonstrates the calculation of a Surface Mass Balance (SMB) field for the Argentière Glacier in the Mont Blanc massif using high-resolution dh/dt fields (from Pleiades DEMs) and the calculation of ice flux divergence (using three different ice thickness datasets, a high-resolution velocity field, and two different methods to calculate the divergence). The resulting SMB field is then compared with results from an SMB model and stake observations, with the ultimate goal of obtaining a precipitation correction that better represents avalanche deposits. The impact of this correction on the glacier's future ice volume is subsequently examined using a 3D thermomechanical ice flow model.

The manuscript is very well-written, with a clear sequence of the steps undertaken by the authors. The reader can easily follow the process, which allows to apply a similar setup for other glaciers (highly appreciated). The various figures, both in the text and in the supplementary material, are well-chosen to represent the results obtained. In addition to the very detailed work carried out (including all uncertainties), the result is compelling and demonstrates the significant added value of the method.

In my opinion, this paper is definitely a valuable contribution to the glaciological community and is therefore highly recommended for publication in 'The Cryosphere.'

We would like to thank Dr Van Tricht for his thorough review and his very relevant and constructive comments.

I have only a few textual comments and a couple of minor clarifications which I would like to see in the revised version. The only more substantial work I can see is calculating/presenting a mass balance derivation for each year in the studied period (2012-2021). How well does the mean specific mass balance obtained with the applied

method represent the mean specific mass balance from the glaciological method and SMB model? And how does the mass balance of the stakes per year compare with the method used?

Thanks for this suggestion. There were several reasons why we choose this relatively long time period to conduct our SMB inversions, which are mainly due to uncertainty considerations:

- When reducing the observation period to time intervals shorter than 3 years it has been shown that at the glacier scale the density conversion factors could deviate outside the range of 850 +/- 60 kg/m$^3$ (Huss, 2013). To stay within reasonable uncertainty bounds, particularly in the accumulation zones, we therefore chose to stick to a longer time series.
- Elevation change and velocity data become very gappy at the annual scale, despite the quality of the Pléiades data. These gaps, especially at the base of the headwalls (which are subject to steep slopes and topographic shadows that affect the velocity and elevation data) need to be filled for our inversion approach and this may lead to artefacts and inconsistencies in the products. Figure S16 show examples of velocity maps for the periods 2012-2015, 2015-2018 and 2018-2021 which indicate a high density of gaps at the base of the headwalls, particularly for the period before 2018. Aggregating these velocity maps is therefore crucial to reduce the uncertainty in the velocity product. Similarly, one can refer to the Figure 2 in Beraud et al. (2023) (see copy below) that indicates the numbers of gaps needing to be interpolated in the Argentière DEMs. While their study also includes winter pairs, this shows that some locations on the glacier are particularly difficult to map, thus the need to use as many DEMs as possible to constrain the elevation change patterns in these locations.

[Figure]

**Fig. 2 from Beraud et al. (2023).** Left: Map of the Glacier d'Argentière showing the number of interpolations between consecutive DEMs (total 11 pairs). Right: Snow-free stable areas used by three co-registrations or more. Spring areas nearly fully cover summer areas.

- We also expect a high temporal variability in accumulation (especially from avalanching, Hynek et al., 2024), as can be seen in the elevation change patterns

shown in Figure 4 of Beraud et al. (2023). While it is definitely very interesting to analyse this annual variability, we were more interested in the average contribution of avalanches to the surface mass balance over a time period long enough to smooth out this variability and analyse the long-term signal.

For all these reasons we decided to stick to our relatively long study period of 2012-2021, which uses all the data that was available from the Pléiades observations. We agree that it could have been possible to reduce this period by a few years or cut it in two without increasing the uncertainties too much, but this was not necessary for the purpose of our research questions and therefore felt beyond the scope of this study. We will therefore keep it as it is. These points are already mentioned in the discussion section (L479-484):

'We computed the mean value for the full 2012-2021 period, but assuming that the flux divergence and firn density remain constant over ~1 decade, these could be refined to yearly or even seasonal time-scales using high-resolution elevation changes from the Pléiades DEMs (Jourdain et al., 2023; Zeller et al., 2023), with some additional uncertainties caused by the DEM differencing over shorter time periods (Beraud et al., 2023).'

Specific comments:

- Line 15: "Particularly" might be removed here

  Removed as suggested (L15)

- Line 16: High resolution (<-> high quality) as well?

  Added as suggested (L16)

- Line 18: A bit unclear if the approach to invert only uses three different ice thickness estimates or three different methods. Further, if formulated like now, it seems that the ice thickness setups show a good agreement, but it is the three different inversions that do show the good agreement.

  Agreed. Changed as follows (L18-21):

  'Three inversions are conducted using three different ice thickness modelling approaches, two of which are constrained by observations. The inversions all show a very good agreement between inverted surface mass balance and *in situ* measurements (RMSE between 0.50 and 0.96 m w.e. yr$^{-1}$ for the 11-year average).'

- Line 19: After reading, the "consensus F2019 estimate" is not constrained by the ice thickness measurements?

  See response to comment above.

- Line 20: Maybe mention the range of RMSE values?

  Added as suggested (L20).

- Line 21: "the" modelling approaches

  Modified as suggested (L21).

- Line 23: "Avalanching" -> avalanche deposits

Modified as suggested (L24).

- Line 33: Accumulation zone

Modified as suggested (L35).

- Line 36: and extrapolation?

Added as suggested (L38).

- Line 41: Compaction as well?

We considered compaction to be an 'internal process', we have added it as an example (L44): 'This mismatch is due to internal processes such as compaction'

- Line 45: Remove "and"

Removed (L47).

- Line 52-53: Potentially add a reference to "Turchaninova A.S., Lazarev A.V., Marchenko E.S., Seliverstov Y.G., Sokratov S.A., Petrakov D.A., Barandun M., Kenzhebaev R., Saks T. Methods of snow avalanche nourishment assessment (on the example of three Tian Shan glaciers). Ice and Snow. 2019;59(4):460-474. https://doi.org/10.15356/2076-6734-2019-4-438

Really nice reference, thanks! Added here (L52-53).

- Line 67: "ice flux"

Modified as suggested (L69).

- Line 67: Therefore -> subsequently

Modified as suggested (L69).

- Line 81: "which" does not have a direct link here. You probably mean that ice thickness and velocity is less constrained in the accumulation zone and therefore the uncertainty of the product is larger, butt this might need to be a bit clearer stated

We modified the sentence for clarity (L82-84): 'However, these estimates depend on the quality of the ice thickness, velocity and elevation change data which are less constrained and therefore lead to higher uncertainties in the accumulation area of the glaciers (Miles et al., 2021).'

- Figure 1: The colour scale is a bit large for the values of the SMB (especially the positive ones). Maybe limit this to +3 m w.e. yr-1 so that you can see a bit more the variations? The circles of the GPR locations are too large so that you cannot locate the individual points (it now seems to be more a line of GPR). Panel b, is it an option to zoom in somehow (e.g., to the Mont Blanc massif)?

Thanks for these suggestions. We tried to use an asymmetrical color scale to constrain it to [0; 3] m w.e. yr-1 for the positive values, however we want to keep the same colorbar throughout the manuscript and if we cap it at 3 m w.e. yr-1 then the very positive values from the SMB inversion saturate. We have therefore kept it as it is. The GPR acquisitions are very dense so they really are more lines than points, we have changed the symbol in the legend. We have also zoomed panel b to the Mt. Blanc massif.

- Line 132: Which software is used to compute the surface displacement, velocity?

  The processing chain used is the same as the one described in Millan et al. (2019) and more recently in Mouginot et al. (2023), which uses Python 3.7 and Fortran. It does not have a particular name. We have specified this in the text (L134-136):

  'The velocity fields were obtained using normalised cross-correlation and the images were co-registered using the median velocity of the off-glacier terrain using the workflow described by Millan et al., (2019) and Mouginot et al. (2023).'

  Mouginot, J., Rabatel, A., Ducasse, E., & Millan, R. (2023). Optimization of Cross Correlation Algorithm for Annual Mapping of Alpine Glacier Flow Velocities; Application to Sentinel-2. *IEEE Transactions on Geoscience and Remote Sensing, 61*, 1–12. https://doi.org/10.1109/TGRS.2022.3223259

- Line 141: You mention later that the F2019 did not use the GPR measurement, so it is a bit strange to mention here that the three methods use in situ data

  We have modified the sentence as follows to clarify this point (L144-145): 'We used distributed ice thicknesses obtained from three different modelling approaches, two of which are constrained by *in situ* ice thickness observations.'

- Line 184: There is some error between ice thickness (H) change (dH/dt) and surface elevation (h) change (dh/dt).

  We have correct this for equations (2) and (3).

- Line 184: Formulating $H_2O$ makes the equation for me a bit confusing. Consider writing just $r_w$ and $r_i$.

  We have changed $rH_2O$ to $r_w$

- Line 185: Any evidence that internal and basal mass balance are negligible from previous studies?

  The good agreement between geodetic mass balance and surface mass balance measurements shown in Beraud et al. (2023) seems to comfort this hypothesis. Basal ablation from geothermal heat is generally very limited on temperate alpine glaciers (Alexander et al., 2011). Neglecting the internal mass balance is a stronger hypothesis, which is discussed in more details in the discussion section (L501-508):

  'In our processing we assumed that there is no significant change in firn compaction rates, and as such the influence of the density uncertainties on the final uncertainty are of secondary concern (Fig. S9, S10). This assumption may not hold for other glaciers and other time periods, for which changing firn densification may lead to surface lowering with little influence on the surface mass balance or flux divergence (Belart et al., 2017; Pelto et al., 2019; Réveillet et al., 2021; Vincent et al., 2020; Zeller et al., 2023). Firn compaction could partly explain the low SMB values relative to the Tour Noir GLACIOCLIM measurements in all three modelling approaches. To give an order of magnitude, assuming a 20 m thick firn layer, a 100 kg m$^{-3}$ increase in mean density over the study period would explain a 0.2 m w.e. yr$^{-1}$ SMB difference, which is however insufficient to explain the differences between observed and SIA or F2019 SMB.'

We have added the two references mentioned above to the sentence to justify this choice (L186-187).

Alexander D, Shulmeister J, Davies T. High basal melting rates within high-precipitation temperate glaciers. *Journal of Glaciology*. 2011;57(205):789-795. doi:10.3189/002214311798043726

- Line 188: Here you refer to ice thickness as H

Apologies for this, we have changed it in equation 3.

- Line 211: Any estimate for y based on observations or modelling?

To our knowledge, no study has focused on distributed estimates of '*y*' for Argentière Glacier, which is why we derived it from the IGM simulations (Fig. S7). There is a preprint in EGUsphere that has looked at ice deformation using borehole data in the lower part of Argentière Glacier (Roldan-Blasco et al., 2024), finding a yearly mean vertical velocity of 38 m/yr for a surface velocity of 43 m/yr, so a 0.88 y ratio. However, these are local measurements in the lower part of the ablation area and likely not representative of the entire glacier. We therefore considered a uniform distribution of y between 0.8 and 1 to quantify its uncertainties in a conservative way (Cuffey & Paterson, 2010).

- Line 215: If formulated like this, it seems you calculate the ice flux divergence in three different ways. But as far as I understand, this is not the case? Three different ice thickness estimates are used, and two different approaches to determine the ice flux divergence.

Yes, good point, we have made this more explicit (L216-217): 'We computed the flux divergence using three different ice thickness estimates and two different approaches, leading to three SMB estimates (Eq. 2), the **SIA, F2019** and, **IGM** estimates.'

- Line 223 and 224: Some repetition here of the ranges for the uncertainty (with lines 208-213).

We have removed this sentence to avoid repetition (L225-226).

- Figure 2: Like for F1, is it possible to limit the range of the colour scale for the SMB?

Please refer to our response for figure 1: we did not go for an asymmetrical color bar to prevent saturation in the distributed SMB figures.

- Line 243: I do not completely get this 10% best SMB estimates. Can you clarify this a bit?

Yes, apologies if that was not clear. The idea here was to take the best 10% scenarios (relative to in situ measurements) to be used as a reference to quantify the spatial variability and contribution from avalanches. We have clarified it at the end of the sentence (L244-247):

'We compared the inverted distributed SMB with the *in situ* SMB at the GLACIOCLIM stake locations and for each modelling approach selected the best 10% SMB estimates that minimised the weighted

quadratic sum of the RMSE of the Argentière and Tour Noir stakes to be used as reference scenarios for the quantification of the avalanche contribution.'

This complements the details given in 2.9 (L298-300):

'Here, we re-evaluated this Pfact over the Rognons, Tour Noir and Améthystes tributaries and in the accumulation area using the mean of the best 10% inverted SMB patterns of each modelling approach'

- Line 273: How does the exponentially decay of the albedo is determined? Which time scale is used?

This exponential decay is based on the parametrization presented in Hock and Holmgren (2005), using the same parameters as in their study, with a decay of -0.1 day-1. We have added this reference here (L276-277):

'Where $\alpha$ is the local surface albedo (=alpha_ice or =alpha_snow, which decreases exponentially with the age of the surface snow; Hock and Holmgren, 2005),'

- Line 341: Again, here you mean dh/dt I guess? The difference between dH/dt and dh/dt is not clear throughout the manuscript. I guess because bedrock elevation is considered to be stable, both are the same, but you should state this somewhere and from then on work always with dH/dt

We have changed this to dh/dt (L344).

- Line 343: The median of the 2012-2021 period for every grid cell?

Correct, we have specified this here (L347).

- Figure 3: Both for the elevation change and the surface velocity, the colour scales could be optimized (wider for velocity, smaller for elevation change).

We have extended the color bar to [0 100] m/yr for velocity. However, as explained above we have kept our colorbar for the SMB to prevent saturation in the zones influenced by avalanches.

- Line 348: Median or mean velocity (<-> line 343)?

Good catch, this is the median, we have corrected it in the caption (L352).

- Figure 5: You state different modelling approaches, but in fact it concerns two different ice flux modelling approaches and three different ice thickness estimates?

For clarity we have changed this sentence to (L386): '(a) SIA, (b) F2019 and (c) IGM SMB estimates.'

- How well can you invert the annual surface mass balance? There is only a focus on the multi-year average

Please refer to our answer to the general comment above.

- Line 385-389: I do not completely get this sentence (which I find too long).

We have cut it into two different sentences for clarity and simplified the second part (L389-393): 'There is a good agreement with the SMB measurements on Argentière for both the SIA and F2019 thickness with RMSE values lower than 0.67 m w.e. yr$^{-1}$ (Fig. 6d-e). Uncertainties

of the F2019 and SIA estimates reach up to +/- 1.2 m w.e. yr$^{-1}$ (F2019) and +/- 0.8 m w.e. yr$^{-1}$ (SIA) at the stake locations and higher at the margins of the glacier, particularly over the Rognons tributary (Fig. S6), where they locally reach up to +/- 6-8 m w.e. yr$^{-1}$'

- Figure 9: Very cool to see the impact of taking into account the avalanches. I wonder, however, if it is possible to show the flowlines. My first guess would be that the ice at the location of the avalanche deposits does not flow to the central trunk but moves along the headwall. But this must be different because the central trunk maintains more ice?

Indeed, according to our simulations with Elmer/Ice, initially the ice at the location of the avalanche deposits does not flow to the central trunk but moves along the headwall (Fig. R1). However, as the glacier retreats the geometry changes and the flow from the avalanche cones does converge into the main glacier trunk.

[Figure]

*Figure R1: Glacier flowlines in 2021 for the corrected scenario*

- Line 558: Is shading not included in the simplified energy balance model? By modifying the incoming radiation?

We did not include shading in our simplified energy balance model to reduce the computation cost. This decision is based on preliminary tests that we conducted and that showed that including shading had a limited effect on the surface mass balance as: 1/ in winter the melt is very limited even in the non-shaded locations and 2/ in summer the sun is high enough in the sky that the large majority of the glacier surface receives the same amount of SW radiation. We do agree however that in some cases topographic shading could lead to strong mass balance gradients, and that it will be worthwhile to account for this effect in glacier models. This was mentioned in the discussion (L561-564):

'These could be related to the topographic shading at the base of these north-facing headwalls, to the preferential redistribution of snow by wind, to a varying precipitation lapse rate at high elevation along

the headwalls, or to the lower albedo values caused by the snow cover lasting longer on the avalanche cones (Florentine et al., 2018; Olson and Rupper, 2019).'

- Line 603: I guess also the ice thickness is crucial when the approach would be applied on larger scales? As you show in the sensitivity analysis

  It is true that in our sensitivity analysis (Fig. S9-S10) ice thickness is responsible for most of the uncertainty. However, this is a particular case as we have very high quality velocity data for our study. In the end all three ice thickness scenarios seem to perform relatively well, but when we changed to a coarser resolution velocity product (Fig. S15) this really had a strong negative impact on the results of the SMB inversion. This is why we have stated that velocity seems to be the main current limitation. But of course, this would need to be tested at other sites. We have now also mentionned ice thickness here (L605-606): 'with the main current limitations being the quality of the surface velocity observations, especially in the accumulation zones, followed by the availability of ice thickness measurements to constrain distributed ice thickness estimates.'

- Line 606: cf line 16 high-resolution vs high-quality (both are true…)

  Good point, we have added 'high quality' here (L608).